# A cooperative nano-CRISPR scaffold potentiates immunotherapy via activation of tumour-intrinsic pyroptosis

Ning Wang [1,2], Chao Liu [1,2], Yingjie Li [1], Dongxue Huang [1], Xinyue Wu [1], Xiaorong Kou [1], Xiye Wang [1], Qinjie Wu [1] ✉ & Changyang Gong [1] ✉

Efficient cancer immunotherapy depends on selective targeting of high bioactivity therapeutic agents to the tumours. However, delivering exogenous medication might prove difficult in clinical practice. Here we report a cooperative Nano-CRISPR scaffold (Nano-CD) that utilizes a specific sgRNA, selected from a functional screen for triggering endogenous GDSME expression, while releasing cisplatin to initiate immunologic cell death. Mechanistically, cascade-amplification of the antitumor immune response is prompted by the adjuvantic properties of the lytic intracellular content and enhanced by the heightened GDSME expression, resulting in pyroptosis and the release of tumor associated antigens. Neither of the single components provide efficient tumour control, while tumor growth is efficiently inhibited in primary and recurrent melanomas due to the combinatorial effect of cisplatin and self-supplied GSDME. Moreover, Nano-CD in combination with checkpoint blockade creates durable immune memory and strong systemic anti-tumor immune response, leading to disease relapse prevention, lung metastasis inhibition and increased survival in mouse melanomas. Taken together, our therapeutic approach utilizes CRISPR-technology to enable cell-intrinsic protein expression for immunotherapy, using GDSME as prototypic immune modulator. This nanoplatform thus can be applied to modulate further immunological processes for therapeutic benefit.

Protein-based therapeutics have attracted increasing attention and shown potential for cancer treatment[1,2]. Particularly, various proteins such as tumor neoantigens, cytokines as well as pyroptotic substrate (Gasdermin family) have been explored for tumor immunotherapy during the last years[3–5]. Despite these prominent potential benefits of these therapeutics, the direct use of the free proteins remains ineffective by several factors that are attributable to unfavorable physiological barriers in vivo[6–8]. Such agents were easily cleared in the blood or metabolic process, resulting in low bioavailability and thus compromise the therapeutic effect[9,10]. Recent evidence suggests tumor accumulation can be ameliorated by increasing the protein dose, but it also causes immune-related adverse events due to the lack of tumor targeting ability of the protein[11]. The abovementioned limitations have led to the need for strategies to improve protein-based therapeutics. One of the main methods is to increase the stability and bioavailability of protein in vivo by functional modification[12,13]. According to the reports, the therapeutic efficacy of these proteins is indeed ameliorated through some sophisticated approaches. An alternative is the development of delivery system, which has also attracted increasing interest. With the assistance of carriers, the protein can be accumulated at the tumor site and taken up by the tumor cells[14–16]. However, prevailing approaches are mainly focusing on exogenous delivery of

[1]State Key Laboratory of Biotherapy and Cancer Center, West China Hospital, Sichuan University, Chengdu 610041, China. [2]These authors contributed equally: Ning Wang, Chao Liu. ✉e-mail: cellwqj@163.com; chygong14@163.com

therapeutic proteins which are costly and laborious. Furthermore, the extremely demanding in terms of bioactivity of proteins also detracts from successfully moving these strategies from the bench to the clinic[17–19].

The utilization of intracellular bioactive proteins without exogenous delivery aims to afford a safe and natural manner to treat cancer[20,21]. Tumor cells are derived from normal cells whose genomic composition contains sequences coding for proteins with therapeutic functions[22]. The mechanism provides an opportunity to take advantage of these biological characteristics to produce proteins with therapeutic effects via a tumor self-supply manner. Since the protein is furnished directly by tumor cells, it is unnecessary to consider the negative impact of the delivery process on protein stability and activity, so as to greatly improve the bioavailability. Notably, self-supplied protein plays the therapeutic function within tumor cells immediately, so the undesirable adverse effects on normal cell are avoided. Unfortunately, as a kind of malignant cells, the functional proteins are usually ectopic expressed (not express or express in low level; for example, tumor suppressor: TP53, pyroptotic substrate: GSDME, ect.) due to genetic mutation or epigenetic silencing[23–26]. Thus far, in the face of these obstacles, only limited efforts have been devoted to this area. Therefore, developing a facile approach for constructing a versatile platform that can efficiently realize the self-supply of the therapeutic proteins intracellularly remains desirable but challenging. By fusing the nuclease-inactivated Cas9 with transcription activators, CRISPR/dCas9 can use the own genome of tumor cells to produce therapeutic proteins with bioactivity, providing a precise and self-supply modality to yield GSDME during pyroptosis process[27–29].

Here we present a cooperative Nano-CRISPR scaffold (Nano-CD), self-supplying GSDME, developed for intracellular pyroptosis-based immunotherapy. Briefly, Nano-CD is fabricated by coating a versatile copolymer on amino acid modified cationic core for CRISPR/dCas9 and cisplatin co-delivery. The well-designed structure endows Nano-CD with stability in blood circulation, as well as on-demand release of cisplatin and CRISPR/dCas9 plasmid in the acidic intracellular environment. Ultimately, owing to the cooperative of GSDME protein from tumor self-supply and cisplatin induced activation of caspase-3, powerful tumor pyroptosis was initiated precisely and efficiently, further reversing the immunosuppressive TME and boosting the antitumor immune cascade as positive feedback. When combined with immune checkpoint blockade therapy, Nano-CD was able to inhibit recurrence and metastasis of malignant melanoma which exhibited strong systemic antitumor immune responses and durable immune memory effect. Given these advantages, our work provides a novel insight into cancer immunotherapy via self-supply of therapeutic protein.

## Results

### Preparation of cooperative Nano-CRISPR scaffold

Nano-CD (the nanoplatform co-loaded cisplatin and CRISPR/dCas9 plasmid) was prepared by first constructing CRISPR/dCas9 polyplex through electrostatic adsorption of amino acid modified PEI and CRISPR/dCas9 plasmid, followed by the formation of the shell by covalent connecting cisplatin and TAT to PEGylated polyacrylic acid (PAA) backbone (Fig. 1, Fig. 2a). To achieve high performance of CRISPR/dCas9 transfection efficiency, amino acids-modified branched PEI 1.8K (Phe/Tyr modified PEI, PEI$_{PT}$) were investigated for construction of the CRISPR/dCas9 polyplex (Supplementary Fig. 1 and Supplementary Table. 1). Meanwhile, the PAA-PEG-cisplatin/TAT copolymer (PCT) was coated onto the polyplex to form Nano-CD with enhanced cellular uptake and the capability of intracellular stimuli-responsive release (Supplementary Fig. 2 and Supplementary Fig. 3). Subsequently, the stimuli-responsive release behavior of cisplatin from Nano-CD was measured by high performance liquid chromatography (HPLC). As shown in Supplementary Figs. 4 and 5, there was

~90% cisplatin released under the simulating lysosome condition (pH 5.0). It is proven that cisplatin can be released in cells accurately and efficiently. Meanwhile, the images observed by transmission electron microscopy (TEM) confirmed the uniform spherical morphology of the nanoparticles (Fig. 2b). Dynamic light scattering (DLS) measurements revealed that the average particle size of Nano-CD under pH 7.4 was $160 \pm 11$ nm and decreased to $116 \pm 5$ nm under pH 5.0, which attributed to the deshelling of protonation PAA backbone in PCT (Fig. 2c). On the other hand, the zeta potential of Nano-CD in pH 7.4 transferred from $-5$ mV to $+14$ mV when exposed to pH 5.0, indicating the responsiveness of Nano-CD under the mimic lysosome condition which facilitated cisplatin and CRISPR/dCas9 plasmid release from lysosomes (Fig. 2d). Furthermore, as illustrated in the gel electrophoresis analysis, Nano-CD achieved DNA condensation successfully with mass ratio of 20: 5: 1, which can efficiently compress and load plasmids to avoid nucleases degradation in vivo (Fig. 2e). The stability of Nano-CD in PBS, 5% serum or pH 4.0-8.0 conditions was shown in Supplementary Figs. 6 and 7.

In addition, the cytotoxicity of Nano-CD components was analyzed by MTT assay. As shown in Supplementary Fig. 8, the PEI$_{PT}$ and PCT had no obvious cytotoxicity even when the drug concentration was 300 μg/mL in B16F10 cells. The intracellular delivery behavior and transfection efficacy of Nano-CD was further studied (Supplementary Fig. 9 and Supplementary Fig. 10). The cell uptake efficiency of Nano-CD increased ~1-fold than that of PEI 25K. Meanwhile, the intracellular behaviors of Nano-CD were explored with a confocal laser scanning microscope (CLSM). The CLSM images indicated that the YOYO-1 labelled Nano-CD (green) began to accumulate to the cell membrane at 1 h. After incubated for 2 h, the green signal trafficked to the lysosome and enriched in lysosome at 4 h. Subsequently, the green signal almost completely overlapped with blue (nucleus) at 8 h, which implied that Nano-CD escaped from lysosome and successfully released the plasmids owing to the layer degradation and "proton sponge" effect of polyethyleneimine (Supplementary Fig. 9c). Benefited from the enhanced cellular uptake and intracellular stimuli-responsive properties, the transfection efficiency of Nano-CD (~80%) in B16F10 cells performing significantly better than that of PEI 25 K (~36%) (Supplementary Fig. 10). The excellent transfection property of Nano-CD scaffold would consequently contribute to the realization of self-supply of protein in tumor cells.

GSDME protein is a key executor during pyroptosis processes because it can be cleaved by caspase-3 into an active membrane perforator domain (GSDME-N)[30–32]. To achieve efficient tumor self-supply of GSMDE protein, we screened the sgRNAs for Nano-CD construction. Among the seven candidate sgRNAs, sgRNA-3 (s3) showed the optimized regulatory efficiency, with a 7.4-fold increase of GSDME expression compared to the untreated group. Therefore, the CRISPR/dCas9 system containing sgRNA-3 was selected for the Nano-CD fabrication. As a result, the relative RNA expression of GSDME in Nano-CD group upregulated 7.5 times compared with blank group (Fig. 2f). Meanwhile, the western blot assay was performed to detect the tumor self-supply of GSMDE protein induced by Nano-CD. As shown in Fig. 2g and h, the protein expression of cleaved GSDME (GSDME-N) and Cleaved Caspase 3 (CC3) was significantly up-related in Nano-CD group, which demonstrated that Nano-CD provided a complete condition for the initiation of pyroptosis. Additionally, the tumor-killing effect were further explored. Compared with the Nano-C (a control nanoplatform loaded with cisplatin and a scramble pDNA) and Nano-dCas9 (a control nanoplatform loaded CRISPR/dCas9 without cisplatin) group which had a puny percentage of cell death (8%), Nano-CD treated cells showed approximately a 5-fold increase of tumor-killing effect (Fig. 2i). These elaborate results implied that cisplatin and dCas9 in Nano-CD do not work alone, but in collaboration to achieve such a superior tumor-killing effect, which consistent with our concept of design.

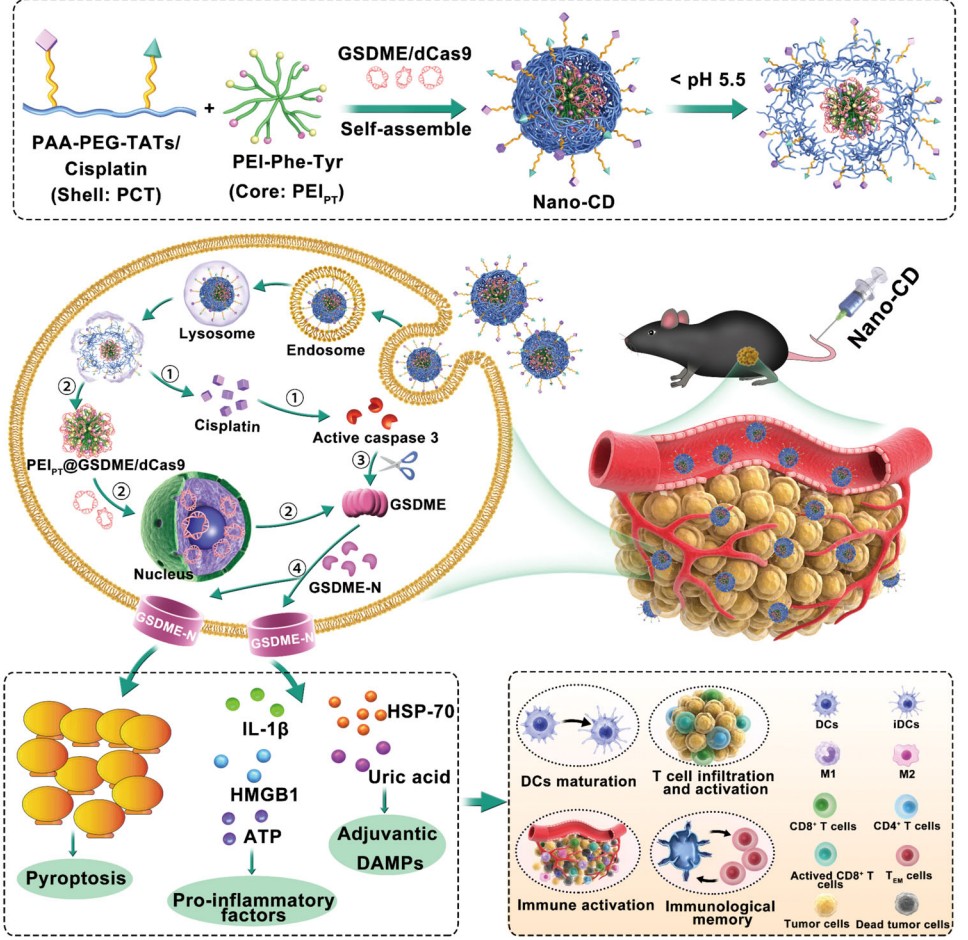

**Fig. 1 | Workflow of cooperative Nano-CD scaffold.** A cooperative Nano-CD scaffold was designed to initiate robust anti-tumor immune responses through self-supply of pyroptotic protein. Upon cellular internalization, Nano-CD protected the components from lysosome degradation and realized hierarchically release of cisplatin and CRISPR/dCas9 plasmid triggered by endosomal acidic conditions. Owing to the modulation of CRISPR/dCas9, endogenous GSDME protein supplied by tumor cells was cleaved by cisplatin induced activated caspase-3 to induce robust pyroptosis, thereby boost efficient cancer-immunity cycle, including DC maturation, antigen presentation, T cell priming and tumor immunogenic death, which demonstrated substantial therapeutic effect of tumor relapse and metastasis in malignant melanoma models when combined with checkpoint blockade.

## Nano-CD-initiated immunogenic pyroptosis

To explore whether the tumor-killing effect was caused by pyroptosis, we characterized the cell death types induced by Nano-CD. We first investigated the morphology of B16F10 cells with different treatments. Nano-CD treated B16F10 cells showed evident swelling with large bubbles from the plasma membrane which is the typical morphology of pyroptosis (Fig. 3a and Supplementary Fig. 11). Based on the pore forming mechanism of pyroptosis[33], the integrity of the cell membrane would be disrupted, followed by the release of cell contents including pro-inflammatory molecules and antigens into the tumor microenvironment. The T11 dye (Red), which could not penetrate the normal cell membrane, successfully stained the proteins in the cytoplasm of the Nano-CD treated cells. The increased red signal implied Nano-CD greatly destroys the integrity of cell membrane by the cooperation of cisplatin and CRISPR/dCas9 (Fig. 3b). Furthermore, the release of cell contents, including adenosine triphosphate (ATP) and lactate dehydrogenase (LDH) in Nano-CD group were strongly higher than that of Nano-C or Nano-dCas9 treated only (Fig. 3c). Meanwhile, the level of IL-1β cytokines, a signal molecule in the classical pyrolytic signaling pathway, also increased significantly in Nano-CD group. A comprehensive look at the above results, we confirmed that Nano-CD initiated robust pyroptosis via the collaboration of self-supply of GSDME protein and cisplatin.

High mobility group 1 (HMGB1) and calreticulin (CRT) are the typical markers of immunogenic cell death (ICD)[34,35]. To further validate the pyroptosis-mediated ICD effect, we studied HMGB1 release and CRT exposure by confocal image analysis, flow cytometry and enzyme-linked immunosorbent assay (ELISA) after treatment with different formulations. B16F10 cells after incubation with Nano-CD induced significant CRT exposure as confirmed by the confocal microscopy. The FCM analysis also exhibited a 5-fold increase in percentage of CRT+ cells after Nano-CD treatment compared to the blank group (Fig. 3d and Supplementary Fig. 12a). Meanwhile, the HMGB1 release was 3-fold increase after incubation with Nano-CD compared with blank group (Fig. 3e and Supplementary Fig. 12b). In addition to the detection of HMGB1 and CRT, we also detected the significant release of adjuvantic danger-associated molecular patterns (DAMPs, uric acid and Hsp-70, HMGB1) in Nano-CD-treated group (Fig. 3f–h)[36]. These data indicated Nano-CD could not only killed the tumor cells, but also reshaped the immune microenvironment by pyroptosis induction via self-supply of GSDME protein, which together guarantees high immunogenicity and self-adjuvant assisted cascade immune activation.

## Dendritic cells (DC) maturation stimulated by Nano-CD mediated pyroptosis

Pyroptosis is a programmed cell death pathway that is critical for antitumor immune activation, which could be induced by caspase-3

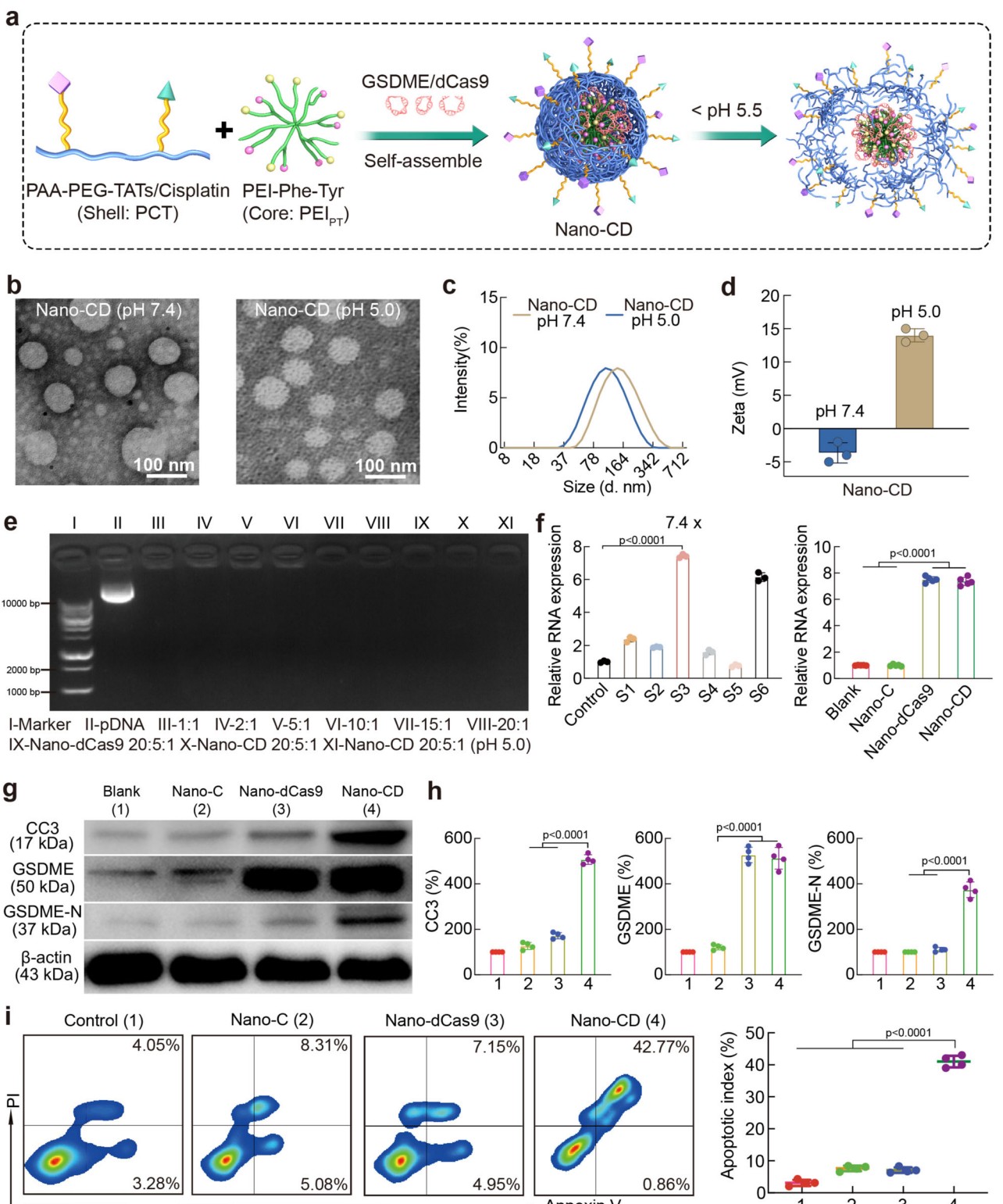

**Fig. 2 | Characterization of Nano-CD nanoparticles. a** Schematic illustration of the preparation of CD particles. **b** TEM images of Nano-CD in pH 7.4 or pH 5.0. **c** Particle size and zeta potential (**d**) of Nano-CD in pH 7.4 or pH 5.0. **e** Agarose gel electrophoresis of pDNA in $PEI_{PT}$ and Nano-CD (Lane 1, DNA ladder; lane 2, naked plasmid; lane 3–8, $PEI_{PT}$@pDNA at mass ratios of 1: 1, 2: 1, 5: 1, 10: 1, 15: 1 and 20: 1, respectively; lane 9, Nano-dCas9 (20: 5: 1); lane 10, Nano-CD (20: 5: 1) and lane 11, Nano-CD (20: 5: 1) in pH 5.0). **f** qPCR analysis for sgRNA screening ($n = 3$ biological replicates per group) and GSDME expression ($n = 5$ biological replicates per group). **g**, **h** CC3, GSDME, GSDME-N, β-actin expression with different treatments by Western Blots assay. **i** FCM assay of apoptosis by Annexin V/PI staining ($n = 4$ biological replicates per group). In panels **f**, **h** and **i**, data are presented as the mean ± s.d. and statistically analyzed using one-way ANOVA and Tukey's tests.

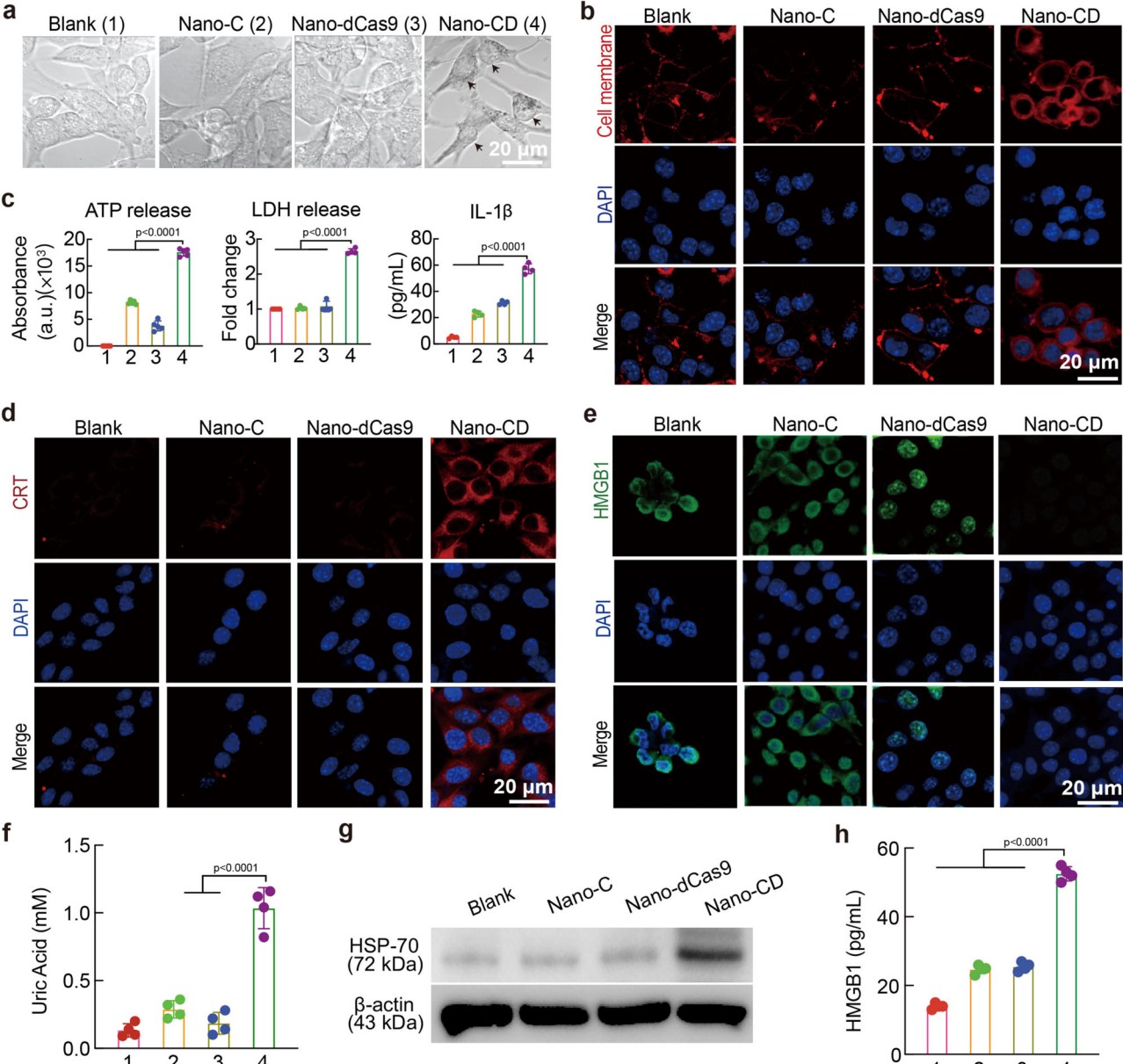

**Fig. 3 | Representative characterization of pyroptosis. a** The morphological images of pyroptosis by CLSM (Scale bar: 10 µm). **b** Observation of ruptured cell membrane by CLSM (cell membrane: red; Nucleus: blue, Scale bar: 20 µm). **c** The extracellular secretion of ATP, LDH and IL-1β. **d** CRT expression analysis by CLSM ($n = 4$ biological replicates per group). **e** CLSM observed the elimination of HMGB1 in B16F10 cells (Scale bar: 20 µm). **f** Quantitive examination of uric acid (n = 4 biological replicates per group). **g** HSP-70 and β-actin expression with different treatments by Western Blots assay. **h** The secretion of HMGB1 in cell supernatant by ELISA ($n = 4$ biological replicates per group). In panels, **c**, **f**, and **h**, data are presented as the mean ± s.d. and statistically analyzed using one-way ANOVA and Tukey's tests.

and bring about systematic inflammation by releasing pro-inflammatory intracellular contents, demonstrating a good opportunity for solid tumor immunotherapy[37,38]. The maturation of dendritic cells can enhance the ability of antigen presentation and the initiation of subsequent antitumor cascade immunity[39]. To explore the immune stimulation of DC induced by pyroptosis via self-supply of GSDME, mouse bone marrow-derived dendritic cells (BMDC) were co-cultured with pre-treated B16F10 cells (Fig. 4a). The DC maturation was induced by Nano-CD mediated tumor pyroptosis (Fig. 4b). The quantitative analysis by ELISA revealed that the tumor necrosis factor-α (TNF-α), IL-12p40 and IFN-γ secretion of mature DC were increased 4.4 -18 times compared with blank group, respectively (Fig. 4c). Furthermore, it was found that the surface expression of CD83 (~40%), CD40 (~42%), CD80 (~71%) and CD86 (~51%), which are well-known markers of DC

maturation, were significantly increased in Nano-CD pre-treated group and were much higher than other groups (Fig. 4d). Comprehensively, Nano-CD treated tumor cells efficiently stimulate DC activation by pyroptosis induction and facilitated in vivo antitumor immune responses.

The capability of tumor targeting is critical to promote the self-supply efficiency of GSDME protein. In order to investigate the biodistribution of Nano-CD, the in vivo & ex vivo tracking was conducted. After intravenous (i.v.) administration, the tumor-bearing mice were imaged at 8 h, 12 h, 24 h, 48 h and 72 h. Nano-CD showed efficient passive accumulation after intravenous injection owing to the enhanced permeability and retention effect (Fig. 4e). No obvious accumulation in major organs included liver, spleen, kidney, heart and lung, which suggested that Nano-CD may

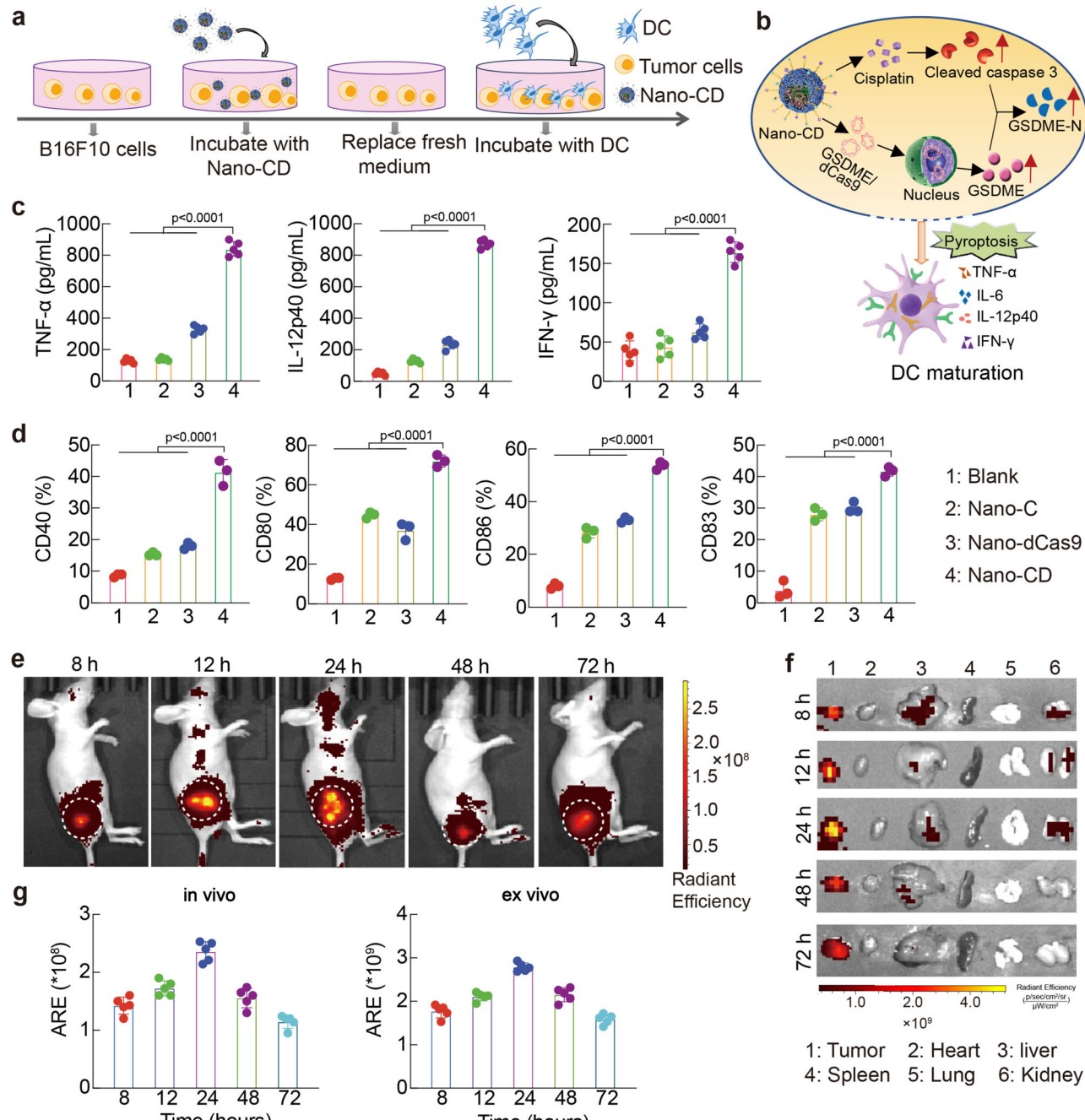

**Fig. 4 | Immune stimulation of DC by Nano-CD in vitro. a** Illustration of the experimental procedure in vitro. **b** The mechanism illustration of DC maturation. **c**, ELISA assay for TNF-α, IL-12p40 and IFN-γ proinflammatory cytokines ($n = 4$ biological replicates per group). **d** Profile of DC maturation by FCM ($n = 3$ biological replicates per group). **e**–**g** Real-time in vivo and ex vivo IVIS imaging of Nano-CD after intravenously post i.v. injection of Nano-CD at 8 h, 12 h, 24 h, 48 h and 72 h. The fluorescence intensity of tumor tissues was also quantified. Data are presented as the mean ± s.d. ($n = 5$ biological replicates per group). In panels **c**, **d**, and **g**, data are presented as the mean ± s.d. and statistically analyzed using one-way ANOVA and Tukey's tests.

realize the full therapeutic potential of protein self-supply approach in vivo (Fig. 4f–g).

## Inhibition of primary tumor growth

As an important negative stimulatory ligand on the surface of T cells, PD-1 could trigger immunosuppressive signaling of adaptive immunity[40]. Encouraged by the above results of efficient tumor-killing and DC activation, we speculated that PD-1 blockade combines with Nano-CD could boost efficient cancer-immunity cycle, including DC maturation, antigen presentation, T cell priming and tumor immunogenic death. Thus, we investigated the anti-tumor efficacy of Nano-CD

and Nano-CD & αPD-1 therapy against malignant melanoma in vivo (Fig. 5a). The mouse melanoma model was first established and treated when tumor volume reached approximately 100 mm³. Nano-CD group presented a higher tumor inhibiting capacity compared with Saline, Nano-C and Nano-dCas9 formulations, respectively. Notably, all the tumor-bearing mice with Nano-CD & αPD-1 treatment showed complete tumor regression, a better result than the other treatments (Fig. 5b and Supplementary Fig. 13). Although there was no significant difference in tumor volume between group Nano-CD & αPD-1 group and Nano-CD group, the survival time of tumor-bearing mice in the combined treatment group were three times than that of Nano-CD

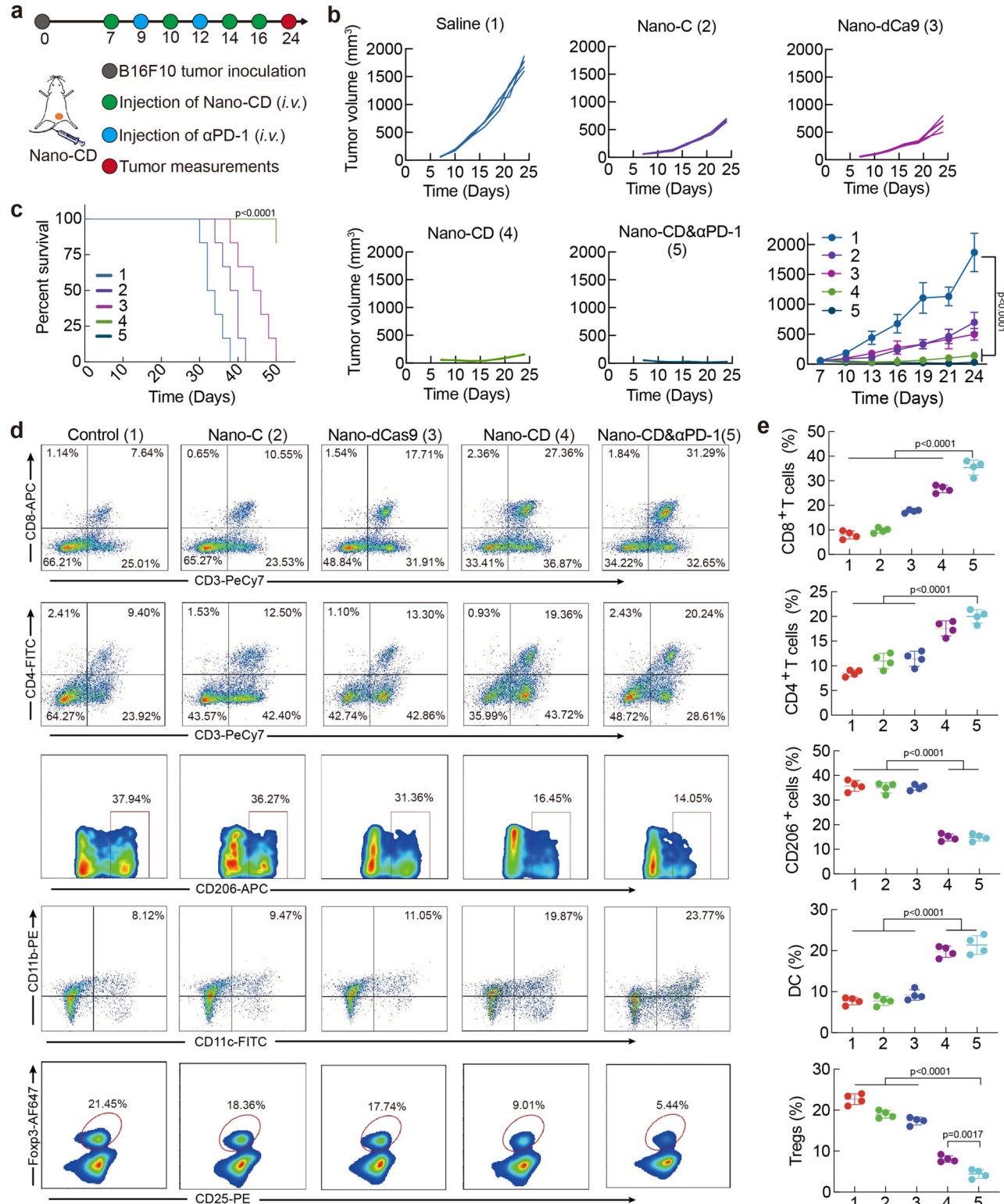

**Fig. 5 | Anti-tumor effect of Nano-CD against primary melanoma model. a** Time schedule of the treatment. **b**, Individual tumor growth curve and average tumor curve during treatment Data are presented as the mean ± s.d. (*n* = 5 biological replicates per group). **c** Survival rate per group. (log-rank test, *n* = 6 biological replicates per group, Nano-CD vs. Saline). **d, e** The percentage of CD8[+] T cells, CD4[+] T cells, CD206[+] macrophages, DC and Foxp3[+] Treg cells in tumors (*n* = 4 biological replicates per group). In panels **b**, **c** and **e**, data are presented as the mean ± s.d. and statistically analyzed using one-way ANOVA and Tukey's tests.

group (Fig. 5c, ****$p < 0.0001$). In addition, the histology examination of tumor tissues illustrated that there were more pyroptotic cells with characteristics including chromatin aggregation, pyknosis and nucleolysis in Nano-CD and Nano-CD & αPD-1 group. Meanwhile, the Nano-CD generally upregulated the GSDME expression in tumors through self-supply manner. The Ki67 and TUNEL staining also confirmed that there was lower cell proliferation and higher cell death ratio in Nano-CD and Nano-CD & αPD-1 than that in other treatments

(Supplementary Fig. 14). Notably, immunofluorescence results of tumor tissues showed a dramatic increase in CD4$^+$ or CD8$^+$ T cell infiltration in the Nano-CD & αPD-1 group, implying the superior performance of immune activation (Supplementary Figs. 15, 16). H&E staining of major organs, complete blood count (CBC test), and blood biochemistry analysis revealed that there was no obvious adverse effect treated with Nano-CD (Supplementary Figs. 17, 18).

To further clarify the underlying mechanism of Nano-CD & αPD-1 treatment, the tumors were collected and the compositions in the TME were further investigated. The percentages of CD8$^+$ cytotoxic T lymphocytes and CD4$^+$ helper T lymphocytes in tumors were observed to be much higher in Nano-CD & αPD-1 (~39% or ~14%) and Nano-CD (~37% or ~10%) groups than that in the other groups (Nano-C: ~17% or ~6%, Nano-dCas9: ~16% or ~7% and Saline: ~10% or ~5%), indicating that Nano-CD could enhance anti-tumor cellular immunity by inducing pyroptosis via cooperation of cisplatin and self-supply GSDME protein (Fig. 5d, e). By contrast, the intratumor infiltration of M2 type macrophages, which could promote the development of tumor[41] were decreased remarkably in mice receiving Nano-CD & αPD-1 (~16%) and Nano-CD (~15%) treatment compared with the other groups (Nano-C: ~36%, Nano-dCas9: ~36% and Saline: ~38%). Moreover, the immunosuppressive T lymphocytes (Tregs) in the tumor microenvironment, which could inhibit the immune response and impede the anti-tumor capability[42], was almost eliminated in Nano-CD & αPD-1 (~5%) and Nano-CD (~9%) compared with the other groups (Nano-C: ~17%, Nano-dCas9: ~18% and Saline: ~21%). Notably, mice treated with Nano-CD & αPD-1 showed significant enrichment of mature DC (~28%) in tumors, which is necessary to initiate cellular immunity. The flow gate strategies of Fig. 5d was shown in Supplementary Figs. 19–23. In addition, the examination of the ICD markers showed that HMGB1 and IL-1β in serum of Nano-CD & αPD-1 and Nano-CD groups was higher than that in Saline group, respectively (Supplementary Fig. 24a, ****$p < 0.0001$). The secretion of IFN-γ and TNF-α in serum associated with immunostimulatory were also significantly increased (Supplementary Fig. 24b). Collectively, all the results indicated that Nano-CD treatment combined with PD-1 antibody could dramatically inhibit primary tumor growth and prolong the survival time. Nano-CD exhibited excellent ICD effect for adaptive immune activation via cisplatin and self-supply GSDME cooperation.

### Inhibition of tumor recurrence

Tumor recurrence, which had high frequency and degree of malignancy, brings great challenges for tumor immunotherapy[43]. Encouraged by the excellent therapeutic efficacy achieved on the primary tumor growth, we further studied the efficacy of the combination therapy to mouse models with recurrence. The majority of the primary tumor was surgically resected and treated with different formulations when the volume of recurrent tumor reached 100 mm$^3$ (Fig. 6a). Compare with the treatments (Saline, Nano-C and Nano-dCas9 groups) which had no therapeutic effect on tumor recurrence, the combination therapy regressed recurrent tumor in the whole period of 60 days without mouse death, demonstrating high performance in inhibition of malignancy recurrence (Fig. 6b–d). Consistent with the above result, H&E staining also demonstrated that Nano-CD & αPD-1 achieved more cell death in tumor tissues (Supplementary Fig. 25).

Subsequently, the immune mechanism was further studied. As shown in Fig. 6e, f and h, Nano-CD & αPD-1 treatment dramatically promoted the infiltration of intra-tumoral CD4$^+$ T (~11%), CD8$^+$ T (~42%) cells. Although Nano-CD & αPD-1 and Nano-CD treatment showed comparable infiltration of CD8$^+$ T cells, the proportion of active CD8$^+$ T (CD8$^+$CD69$^+$) cells in the combination therapy group (~13%) was 11.5-fold higher than that of the PBS group, and 1.5-fold higher than that of the Nano-CD group, further demonstrated the key role of PD-1 blockade for CTL activation in vivo. Moreover, the effector memory T cells (TEM, CD44$^+$CD62L$^-$) (~91%) in TME increased significantly in Nano-CD

& αPD-1 group, which is essential for preventing tumor recurrence via recognition of "old antigens". Excitingly, we found that that the proportion of CD4$^+$ T, CD8$^+$ T cells in the spleen also increased, indicating that the systemic immunity was activated after the combination therapy (Fig. 6g, i). The flow gate strategies of Fig. 4d was shown in Supplementary Figs. 26–28. These results demonstrated that the great promise combining Nano-CD treatment with PD-1 blockade to surmount tumor recurrence, a major challenge in clinical cancer treatment. In addition, the level of IL-12, IL-10 and TGF-β in serum increased to 15-fold, 10-fold and 6-fold when treated by Nano-CD & αPD-1than that of PBS group, respectively (Fig. 6j). The upregulated secretion of these cytokines could reverse the immunosuppressive TME and in turn amplify the adaptive antitumor immunity.

### Inhibition of tumor distal metastasis

Tumor distal metastasis remain the main cause of clinical treatment failure[44]. Inspired by the systemic immune activation during the tumor recurrence treatment, we next explored the potential of the therapeutics for preventing lung metastasis of malignant melanoma (the common cases in advanced melanoma) (Fig. 7a). B16F10 cells were intravenously injected into the tumor-bearing C57BL/6 mice to establish the lung metastasis model. Nano-C and Nano-dCas9 showed inefficacy to inhibit the lung metastasis of melanoma. Significantly, Nano-CD & αPD-1 significantly suppressed the lung metastasis. The metastasis nodus in Nano-CD & αPD-1 group was 50% lower than that in the PBS group. (Fig. 7b and Supplementary Fig. 29). In addition, the survival benefit in Nano-CD & αPD-1 group was prolonged to 31 d, that was 1.3-fold longer than the PBS group (Fig. 7c). The H&E staining of lung tissues also illustrated that Nano-CD & αPD-1 treatment exhibited significantly less metastatic tumor lesions than that of other treatments (Fig. 7d). The results indicated that treatment of Nano-CD mediated pyroptosis combined with PD-1 antibody would efficiently inhibit the progression of lung metastasis of melanoma and provides survival benefit to mice with advanced lung metastasis.

To understand the underlying mechanism of this therapeutic efficacy, the systemic immune responses, as well as major cytokines in serum was further studied. We found that there was more CD8$^+$ cytotoxic T lymphocytes and CD4$^+$ helper T lymphocytes in Nano-CD & αPD-1 and Nano-CD group. Notably, the number of activated T lymphocytes (CD8$^+$CD69$^+$) was remarkably increased in Nano-CD & αPD-1 compared with the other groups (Fig. 7e, f). The flow gate strategies of Fig. 7e was shown in Supplementary Fig. 30. Consistently, the TGF-β, IL-10, IL-12 and IFN-γ inflammatory cytokines in serum, which were important for adaptive antitumor response, were significantly increased (Fig. 7g). The results demonstrated that the robust activation of systemic immune responses triggered by Nano-CD could efficiently inhibit the development of lung metastasis of aggressive melanoma.

## Discussion

In conclusion, we constructed a cooperative Nano-CRISPR scaffold to trigger robust anti-tumor immunity via self-supply modality of bioactive protein. This platform directly utilizes the tumor's self-produced bioactive protein to initiate cell pyroptosis and thereby activate adaptive anti-tumor immunity, without the need for external delivery of active proteins or complex modifications. Benefit from the high performance of in vivo delivery, including prolonged circulation capacity, precise tumor accumulation, enhanced cellular uptake and endosomal escape ability, Nano-CD specifically unlock the expression of GSDME and activate caspase-3 pathway within tumor cells, leading to robust pyroptosis. The immunogenic death and self-adjuvant effect induced by pyroptosis reversed the immunosuppressive microenvironment of tumor and amplified the adaptive antitumor immune cascade. As demonstrated in subcutaneous tumor models and tumor bearing mice with relapse, Nano-CD combined with immune checkpoint blockade exhibited significant inhibition of tumor relapse that

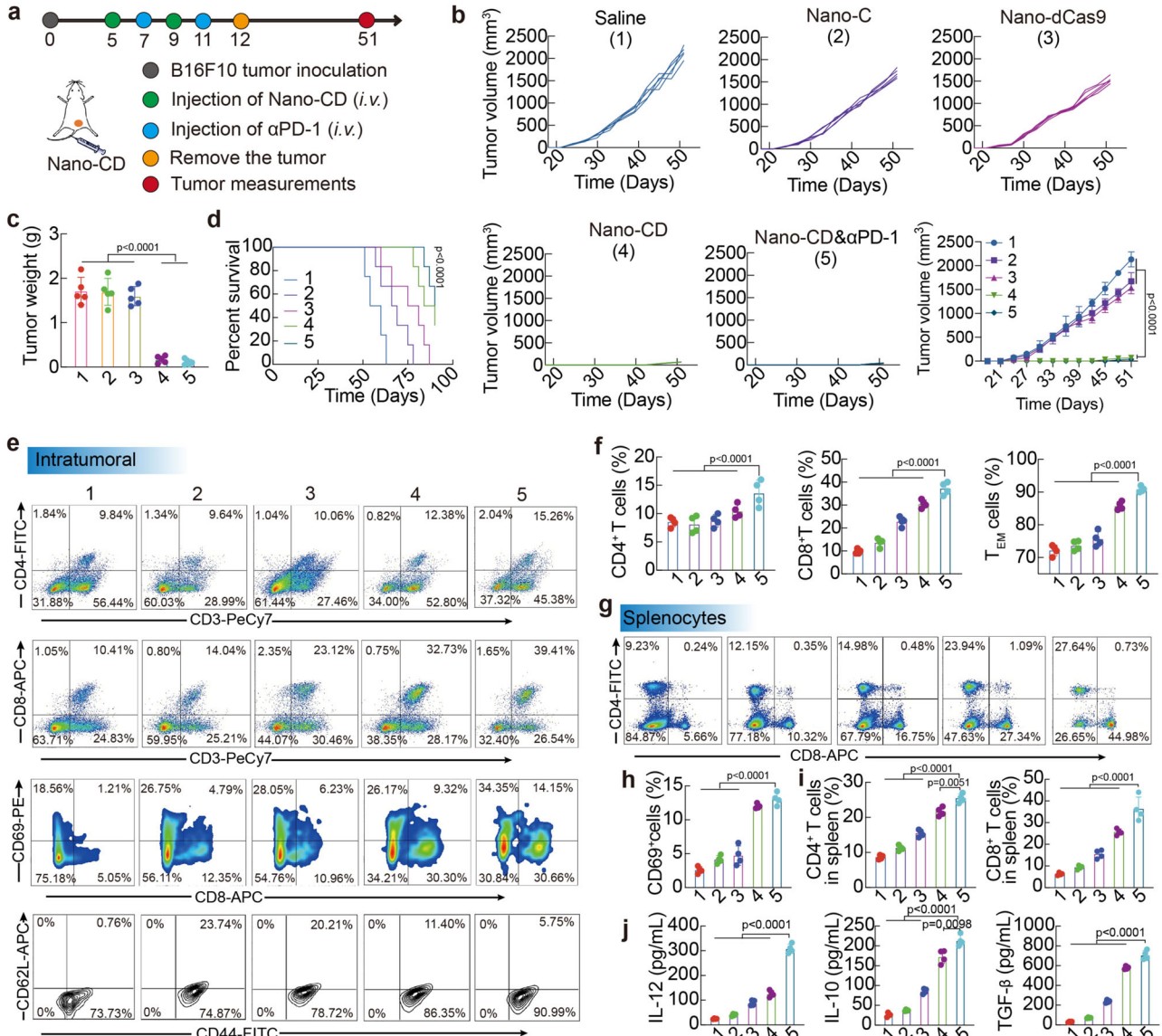

**Fig. 6 | Anti-tumor effect of Nano-CD against recurrence melanoma model. a** Scheme of mice treated with different formulations. **b** Tumor volume (*n* = 5 biological replicates per group), tumor weight (*n* = 5 biological replicates per group) (**c**) and survival curve (log-rank test, *n* = 6 biological replicates per group, Nano-CD vs. Saline) (**d**) of tumor-bearing mice with different treatments. **e**–**i** Profiles and percentages of CD4+ T cells, CD8+ T cells, CD69+ activated T cells,

CD44+CD62L− $T_{EM}$ cells in tumors and CD4+ T cells/CD8+ T cells in spleens by FCM analysis (*n* = 4 biological replicates per group). **j** Concentration of IL-12, IL-10 and TGF-β cytokines in serum after different treatments (*n* = 4 biological replicates per group). In panels **b**, **c**, **f**, **h**, **i** and **j**, data are presented as the mean ± s.d. and statistically analyzed using one-way ANOVA and Tukey's tests.

cannot be completely removed by surgery. Moreover, a strong systemic antitumor immune activation and immune memory effect were observed to protect mice from tumor metastasis. This work provided a new avenue to regulate tumor pyroptosis precisely by chemotherapeutics and CRISPRa plasmid which may have great potential in cancer immunotherapy. Beyond the treatment against malignant melanoma, such nanoplatform may be further extended to treatment of other malignancies that are resistant to conventional therapeutics.

In this study, we focus on the self-supply of the pyroptotic protein GSDME to achieve efficient immunotherapy. This self-sufficient modality may be broadly applicable to a wide range of other proteins, such as tumor antigens, cytokines, chemokines, antibodies that are important for immune activation[45]. While largely improving the performance and bioavailability, the strategy overcomes tedious workflow of conventional preparation and the bioactivity requirement of the bioactive protein. The self-supplied protein acts directly on tumor cells and thus

minimizes immune-related adverse effects. Such a cunning Nano-CRISPR platform may have great value in protein-based cancer immunotherapy for clinical translation.

Due to the challenges presented by direct delivery, current approaches were primarily relied on the delivery of protein-encoding nucleic acids (mRNA or DNA) for intracellular protein expression[46,47]. Unfortunately, the exogenous delivery manner is still limited by the size of the coding sequence and lack of functional isoform, which can only express proteins with small molecular weight[48]. We provide a way to achieve protein self-supply utilizing the genome of tumor cells, theoretically surmounting the limitations of protein size. Prospectively, this nanoplatform with appropriate update can also potentially be employed in the treatment of some other diseases of the kind where the needed functional proteins are particularly large size (e.g., DMD protein, a potential target for Duchene muscular dystrophy)[49].

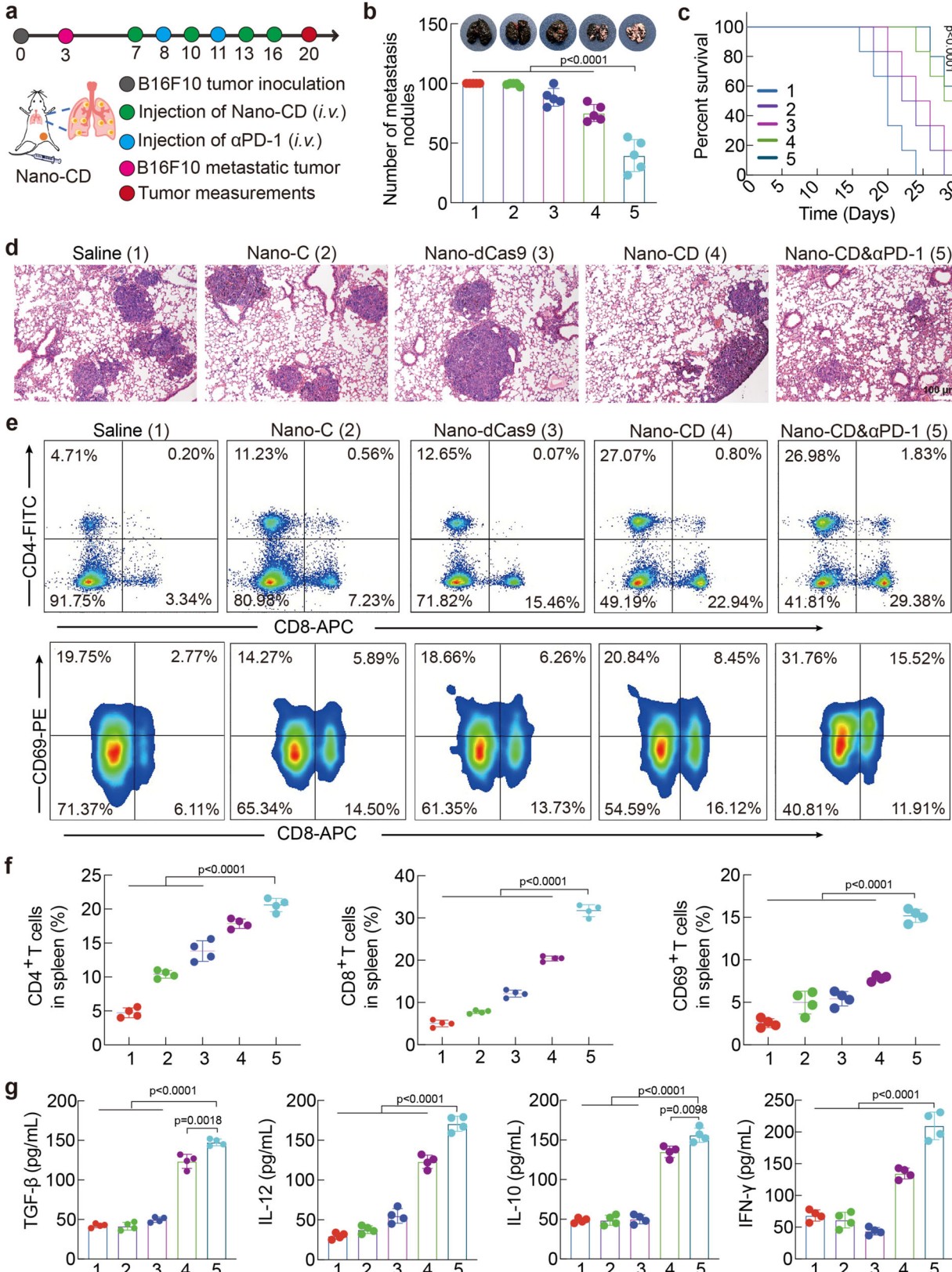

**Fig. 7 | Assessment of therapeutic efficacy of Nano-CD in melanoma pulmonary metastasis model. a** Schematic illustration of administration schedule in vivo. **b** The number of nodules on the lungs was counted and the lungs from each group was pictured at day 20 (*n* = 5 biological replicates per group). **c** Survival curve of mice with different treatments. (log-rank test, *n* = 6 biological replicates per group, Nano-CD vs. Saline). **d** Representative images of lungs at the end of the treatment by H&E assay. **e**, **f**, CD4⁺ T cells/CD8⁺ T cells and CD69⁺ activated T cells in spleens of each group (*n* = 4 biological replicates per group). **g** Level of TGF-β, IL-10, IL-12 and IFN-γ cytokines in serum per group (*n* = 4 biological replicates per group). In panels **b**, **c**, **f**, and **g**, data are presented as the mean ± s.d. and statistically analyzed using one-way ANOVA and Tukey's tests.

# Methods

## Materials

Cisplatin was purchased from Merck KGaA (Germany). $PEG_{600}$ was purchased from ToYong Bio Tech (Shanghai, China). YGRKKRRQRRRC (TATs) polypeptides were synthesized by ChinaPeptides (Shanghai, China). Poly(acrylic acid) was purchased from Sigma-Aldrich (USA). All the other chemical reagents were purchased from Macklin (China). Antibodies for flow cytometry were purchased from Biolegend (Beijing, China). Annexin V-FITC Apoptosis Detection Kit was purchased from Beijing 4A Biotech Co., Ltd. (Beijing, China). Uric acid, HMGB1, IL-1β, IL-12p40, IL-10, IL-12, TNF-α, IFN-γ, TGF-β ELISA kits were purchased from Dakewe Biotech Co., Ltd. (Beijing, China). MTT was purchased from Sigma-Aldrich (USA). DAPI was purchased from Beyotime Biotechnology (Beijing, China). LysoTracker Red, YOYO-1 and TOTO-3 were purchased from Invitrogen (USA).

## Cell lines

Mouse melanoma cell line (B16F10), human melanoma cell line (A375) and human renal epithelial cell line (293T) were obtained from American type culture collection (ATCC). The fetal bovine serum (FBS), Trypsin-EDTA, Trypsin, Iscove's Modified Dulbecco's Medium (RPMI, 1640) and Dulbecco's Modified Eagle Medium (DMEM), Penicillin and cell culture freezing medium were purchased from Thermo Fisher Scientific Co. Ltd (USA).

## Animals

Female C57BL6 (stock#000664) and female Balb/c nude (stock#000711) mice aged 5–6 weeks (14–16 g) were purchased from Beijing HFK Bioscience Co., Ltd. (Beijing, China). The mice were kept in SPF grade animal feeding room and fed with animal food and sterile water. The room maintained properly ventilated and certain humidity at 25 °C. All animal procedures were approved by the Institutional Animal Care and Treatment Committee of Sichuan University (Chengdu, China).

## Synthesis of $PEI_{PT}$ and PCT

Phenylalanine (43.0 mg, Phe), tyrosine (51 mg, Tyr), EDCI (80.0 mg) and NHS (75.0 mg) was added to the MES buffer (pH 5.5, 10 mL) and stirred for 4 h at room temperature. Then, the polyethyleneimine (1.8 kDa, PEI 1.8 K, 100.0 mg) and triethylamine (20 μL) was added into the solution for another 72 h. After dialyzed against ddH2O for 72 h, the $PEI_{PT}$ powder was obtained by lyophilization. The products were confirmed by $^1H$ NMR spectrum with $D_2O$ solvent. The number of Phe and Tyr on the PEI 1.8 K were determined by a ninhydrin assay.

Polyethylene glycol (PEG, 48.0 mg, 0.08 mmoL), EDCI (11.5 mg, 0.06 mmoL) and NHS (6.9 mg, 0.06 mmoL) were dissolved in deionized water under room temperature. After stirring for 0.5 h, TATs (66.4 mg, 0.04 mmol) and 20 μL triethylamine were added into the solution for another 24 h. Subsequently, cisplatin (24.0 mg, 0.08 mmol) dissolved in deionized water was added to the solution for another 24 h. The Boc on PEG were removed for another 2 h. Polyacrylic acid (PAA, 15.0 mg, 0.005 mmoL), EDCI (11.5 mg, 0.06 mmoL) and NHS (6.9 mg, 0.06 mmoL) were dissolved in deionized water under room temperature. After stirring for 0.5 h, the PEG-TATs or PEG-cisplatin were added into the solution for another 24 h. After dialysis in deionized water, PCT was obtained by lyophilization. The structure of the resultants was confirmed by $^1H$ NMR (Bruker, Swizerland) and Fourier transform infrared reflection (FT-IR) spectrum. The grafting amount of cisplatin, and the cisplatin release behavior on PCT in pH 5.5 solution were determined by HPLC. The polymer of shell without cisplatin (PT) was synthesized according to the same methods of PCT. The PT was used as control.

## Preparation and characterization of Nano-CD

Nano-CD nanoparticles were formed by two steps: Firstly, the $PEI_{PT}$@pDNA was assembled by positive charge of $PEI_{PT}$ and negative charge of pDNA at the mass ratio of 5: 1 ($PEI_{PT}$: pDNA), following with gentle mixture and incubated for 30 min at room temperature. Secondly, PCT was added into the $PEI_{PT}$@pDNA, and incubated for another 30 min. Finally, the Nano-CD was used for further experiments. The pH responsive behavior of PCT was also conducted to pH 5.0. The dynamic diameters and zeta potential of $PEI_{PT}$@pDNA and Nano-CD in pH 7.4 or Nano-CD in pH 5.0 was measured by dynamic light scattering nanosizer (Malvern Nano-ZS90, UK). The morphology of $PEI_{PT}$@pDNA, Nano-CD in pH 7.4 or Nano-CD in pH 5.0 was also observed by transmission electron microscopy (TEM, JEOL JEM-100CX, Japan).

The pDNA loading capacity of $PEI_{PT}$@pDNA and Nano-CD was characterized by agarose gel retardation analysis. $PEI_{PT}$ and pDNA was mixed with different mass ratio (1: 1 ~20: 1), and Nano-dCas9 (PT: $PEI_{PT}$: pDNA = 20: 5: 1), Nano-CD (PCT: $PEI_{PT}$: pDNA = 20: 5: 1) and Nano-CD (PCT: $PEI_{PT}$: pDNA = 20: 5: 1) in pH 5.0 were also formed to subject to the gel. After running for 30 min at 140 V, the DNA condensation ability of $PEI_{PT}$ and Nano-CD were detected by the UV transilluminator (BD, USA).

## Detection of cellular uptake and cellular transfection

Generally, the B16F10 cells were seeded in 6-well plates at a density of 1 × 10^5 cells per well and cultured overnight. The pCRISPRa plasmid was incubated with YOYO-1 dye (10,000 ×, green) for 30 min. Then, the Nano-CD @pCRISPRa-YOYO-1 was self-assembling with PCT, $PEI_{PT}$ and pCRISPRa-YOYO-1 at a mass ratio of 20: 5: 1 (2 μg pCRISPRa plasmid per well). The YOYO-1-labeled Nano-CD and other formulations (EGFP-YOYO-1 or $PEI_{PT}$@pCRISPRa-YOYO-1) were added into B16F10 cells. After post-incubation for 2 h, the cell supernatant was removed, and the cells were collected and washed with PBS for twice. The cellular uptake efficiency was determined with the fluorescence intensity by FCM (BD, USA).

Moreover, the intracellular distribution of Nano-CD was observed by CLSM (ZEISS, Germany). Briefly, the B16F10 cells were cultured in glass bottom dishes at density of 1 × 10^5 cells per dish, and then incubated with Nano-CD@pCRISPRa-YOYO-1. After post-incubated for 1 h, 2 h, 4 h and 8 h, the cells were washed with PBS for twice and bound with LysoTracker Red dye (5000 ×) for 1 h. Then, the medium was removed and washed for three times. Subsequently, 5 mL methanol was added into the cells for 15 min. After washed with PBS for twice, the cell nucleus was stained with DAPI for 10 min. Finally, the images were captured by CLSM.

The transfection efficiency of Nano-CD was further determined. The B16F10 cells were seeded in 6-well plate at density of 1 × 10^5 cells per well and cultured for 12 h. After removed by DMEM medium (contained 0% FBS), the PEI 25 K@pEGFP, $PEI_{PT}$@pEGFP and Nano-CD@pEGFP was added into the cells and incubated for 6 h. The medium was gently replaced with DMEM medium (10% FBS), and the cells were cultured for another 48 h. The green fluorescence pictures and fluorescence intensity was analyzed by fluorescence microscope (Olympus) and FCM, respectively.

## MTT assay

The cytotoxicity was measured by MTT assay. The B16F10 cells were seeded in 96-well plate and cultured at a density of 3000 cells per well. After cultured for 12 h, PEI 25 K, PEI 1.8 K, $PEI_{PT}$, PAA, PT or PCT at different concentration (0 to 300 μg/mL) were added into cells for 48 h. Subsequently, the 20 μL MTT solution (0.5 mg/mL) was added. After incubated with cells for 4 h, the supernatants were removed, and the formazan crystals were dissolved in DMSO for 10 min. The OD values were measured by a microplate reader under 570 nm wavelength.

## qPCR detection

To screen the effective sgRNA of GSDME for the following application, there was six sgRNA sequences (s1: 5′ -GGACCGGCTGAGGCATCCAG-3′;

s2: 5′-GAGCCAGAGACTGACCCGGA-3′; s3: 5′-CAGAGACTGACCCGGAC GGT-3′; s4: 5′-CCGGTCCACAAGCTAAGGGA-3′; s5: 5′-TCAGCCGGTCC ACAAGCTAA-3′; s6: 5′-TGGACCGGCTGAGGCATCCA-3′) were designed for further usage. The efficiency of GSDME overexpression by Nano-CD was further investigated by qPCR. The B16F10 cells were treated with Nano-C, Nano-dCas9 and Nano-CD for 48 h. Then, the RNA extraction (RNA Extraction Kit, TIANGEN, China) and qPCR analysis (QIAGEN, USA.) were conducted according to the protocol. The relative RNA expression of GSDME was determined by qPCR (BioRad, USA).

## Analysis of tumor-killing effect

For anti-tumor efficiency assay, the B16F10 cells were seeded into 6-well plates at a density of $1 \times 10^5$ cells per well, and exposed to Nano-C, Nano-dCas9 and Nano-CD for 48 h. The cells were washed with PBS for three times. Some of the cells were stained with Annexin V-FITC for 15 min and PI for another 5 min according to the manual's instructions. The apoptosis of B16F10 was detected by FCM. Additionally, some of the cells were used for protein extraction. Briefly, the cell supernatant was removed, and the cells were collected in RIPA solutions (5 ×). After being lysis for 30 min, the proteins were isolated by centrifugation at 15000 rpm for 15 min. The expression of Cleaved Caspase 3 (CC3), GSDME-N and β-actin (Abcam, USA, 1000 ×) was individually examined by Western Blots assay, respectively.

## Assessing the specificity of pyroptosis

After being treated with Nano-C, Nano-dCas9 and Nano-CD for 48 h, the pyroptosis characteristics of B16F10 cells were visualized by CLSM. The incomplete of cell membrane caused by pyroptosis was stained with T11 dyes which could sign the ruptured cell membrane and isolate from normal cell membrane. After incubated with DAPI for 10 min, the ruptured cell membrane was observed by CLSM.

The extracellular secretion of ATP, LDH and IL-1β was detected by ATP assay Kit (Promega, USA), LDH assay Kit (Promega, USA) and ELISA assay (Invitrogen, USA), respectively, according to the protocol.

HMGB1 elimination from nucleus was verified by CLSM and detected by FCM, respectively. Briefly, The B16F10 cells were cultured in glass bottom dishes at density of $1 \times 10^5$ cells per dish and treated with Nano-C, Nano-dCas9 and Nano-CD for 48 h. After washed with PBS for twice, the cells were fixed with 4% paraformaldehyde. Then, the cells were stained with FITC-conjugated anti-HMGB1 antibody (1000 ×) for 6 h. After washed with PBS, the nucleus was stained with DAPI (100 ×) for 10 min. Finally, the HMGB1 was observed by CLSM. The quantitative expression of HMGB1 was also analyzed by FCM. The HMGB1 in supernatant was further detected by ELISA assay.

CRT expression on the surface of B16F10 cells was observed by CLSM and analyzed by FCM, respectively. The B16F10 cells were cultured in glass bottom dishes at density of $1 \times 10^5$ cells per dish and treated with different formulations for 48 h. After washed with PBS for twice, the cells were fixed with 4% paraformaldehyde for 15 min. Then, the cells were washed with PBS and stained with Rhodamine-conjugated anti-CRT antibody (1000 ×) for 2 h. Furthermore, the cells were then stained with DAPI (100 ×) for another 10 min. Finally, the CRT on cell surface were visualized by CLSM. Then, quantitative analysis of the fluorescence intensity of the CRT were determined by FCM.

As we know, the uric acid and HSP-70 are the main molecules which could produce adjuvant effect. The cells were treated with Nano-CD for 48 h, and the supernatants were collected for the detection of uric acid by ELISA according to the manual. Simultaneously, the HSP-70 (Abcam, USA, 1000 ×) and the horseradish peroxidase (HRP)-labeled goat anti-mouse (10000 ×) secondary antibody in supernatants were also measured by Western Blots assay.

## In vitro stimulation of DC

To investigate the DC maturation, the DC were extracted from the bone marrow of female C57BL6 mice (5-week), and then cultured in medium for 1 week in vitro. The B16F10 cells were treated with Nano-C, Nano-dCas9 and Nano-CD for 48 h, and then incubated with DC for another 48 h. The anti-PE-CD40, anti-PE-CD80, anti-APC-CD86, and anti- PeCy7-CD83 antibodies (100 ×) were used to label the DC and analyzed by FCM. The cell supernatant samples were also collected for the detection of pro-inflammatory cytokines (TNF-α, IL-12p40, and IFN-γ) by ELISA.

## In vivo imaging

Female Balb/c nude mice were injected subcutaneously into the right flank with A375 cells ($1 \times 10^7$ cells in PBS per mice). After the tumor growth to 400-500 mm3, the mice were divided into five groups ($n = 3$ mice per group) for investigating the in vivo biodistribution of Nano-CD. Briefly, pCRISPRa plasmid was labelled with TOTO-3 (10000 ×) for 60 min, and then Nano-CD was prepared by self-assembly of electro-static adsorption. The mass ratio of PCT: PEI$_{PT}$: pCRISPRa-TOTO-3 = 20: 5: 1 (5 μg pCRISPRa plasmid per mice). The A375-tumor bearing mice were intravenously injected with Nano-CD and then euthanized by carbon dioxide inhalation at different time points (8 h, 12 h, 24 h, 48 h and 72 h). The amount of Nano-CD distributed in tumors and different organs (heart, liver, spleen, lung and kidney) were imaged by the IVIS Lumina imaging system (Caliper, USA), and the average radiant efficiency was quantitated by Graph Pad 8.0.

## Explore the anti-tumor effect against subcutaneous xenograft melanoma and immune mechanism

Female C57BL6 mice were used to establish the subcutaneous xeno-graft model. ~ $5 \times 10^5$ B16F10 cells in PBS were injected into the right flank of mice. When the tumor volume reached to 100 mm³, the mice were randomly divided into five groups (Saline, Nano-C, Nano-dCas9, Nano-CD, Nano-CD & αPD-1) (CRISPRa: 5 μg per mice; the mass ratio of PCT: PEI$_{PT}$: CRISPRa = 20: 5: 1, 100 μL). The different agents were intravenously injected into the tumor-bearing mice for evaluation of anti-tumor effect ($n = 5$ each). The administrations were performed intravenously every three days. The weight of mice was recorded, and tumor volume was calculated by the formula (Length × Width²/2) every three days. The mice were considered death when the tumour volume reached 2000 mm³ according to the animal ethics of our institute and animal welfare. The maximal tumor burden permitted by the ethics committee was the diameter of 2 cm, which was not exceeded in this study. The tumor growth curve of per mice and average tumor volume of different groups were draw. The mice were euthanized by carbon dioxide inhalation at day 24, and the tumors and organs were harvested and fixed in 4% paraformaldehyde for H&E staining and IHC immunohistochemistry analysis. The tumor sections were detected with rabbit anti-GSDME (50 ×) and rabbit anti-Ki67 antibodies (50 ×) at 4 °C overnight, and horseradish peroxidase (HRP) conjugated second antibodies (1000 ×) for 40 min. The TUNEL staining assay was performed according to the instructions (BD, USA). The expression of proteins was pictured by fluorescence microscope (Olympus), and the quantitative analysis of the proteins were randomly measured by Image J software.

The serum samples were subjected to Heska Element HT5 Hematology Analyzer and Heska Element DC Chemistry Analyze for safety evaluation. Another animal study was conducted for the observation of survival.

For immune activation assay, the tumor-bearing mice were administrated for three times with different formulations, and the tumors were collected for the preparation of single cell suspension. After filtration through sterile screen (70 μm), the cell suspensions with cell density of $1 \times 10^5$ /100 μL was stained with anti-PECy7-CD3, anti-FTTC-CD4, anti-APC-CD8, anti-FITC-CD11b, anti-PE-F4/80, anti-APC-CD206, and anti-PE-CD11c (100 ×) for 30 min at 4 °C. The CD25 and Foxp3 antibodies were stained according to the operation manual. After washed with PBS for twice, the components of tumor immune

microenvironment per groups were measured by FCM. The serum samples of different groups were subjected to detect the secretion of HMGB1, IL-1β, IFN-γ and TNF-α cytokines by ELISA assay.

### Investigate the anti-tumor effect against recurrent xenograft melanoma

The subcutaneous xenograft melanoma model was set up by subcutaneously injection of $5 \times 10^5$ B16F10 cells on the right flank of mice. 7 days later, the tumors were grown up to ~100 mm³, and the mice were randomly divided into 5 groups as above mentioned ($n = 5$ each). After intravenously administration for 12 days, the tumors were removed by nanoparticles or surgery. The weight of mice, tumor growth and survival time of the mice continued to be monitored. After observation for 51 days, the mice were euthanized by carbon dioxide inhalation and the tumors were harvested, and then fixed into 4% paraformaldehyde for H&E staining.

After post-inoculation of Nano-CD for day 30, the tumors and spleens of the recurrent xenograft melanoma bearing mice were evaluated for immunologic memory effect with Nano-CD by FCM. The single tumor cell suspensions per group were obtained as above mentioned. The samples were further stained with anti-PercpCy5.5-CD3, anti-FTTC-CD4, anti-APC-CD8, anti-PE-CD69, anti-APC-CD44, anti-PE-CD62L antibodies (100 ×). Meanwhile, the single cell suspensions of spleens from each group were also stained with anti-FTTC-CD4 and anti-APC-CD8 antibodies (100 ×). The level of IL-12, IL-10, TGF-β cytokines in serum were also determined by ELISA assay.

### Evaluate the anti-tumor effect against pulmonary metastasis of melanoma

The pulmonary metastasis model of melanoma was established on the subcutaneous xenograft melanoma-bearing mice. Firstly, the mice were injected subcutaneously with $5 \times 10^5$ B16F10 cells at day 0. Three days later, the $1 \times 10^6$ B16F10 cells were intravenously injected into the mice. On day 7, the mice were randomly divided into 5 groups as above mentioned ($n = 5$) and intravenously every three days. After treated for 20 days (four times), the mice were euthanized by carbon dioxide inhalation and the lungs were collected and weighted, and then fixed into 4% paraformaldehyde for H&E staining. Additionally, another treatment on tumor-bearing mice were performed for survival prolong assay.

After post-formulated by different agents for three times, the pulmonary metastasis of melanoma-bearing mice was subjected to the systemic immune response analysis. The spleens of each group were collected, and the single-cell suspensions were filtrated through sterile screen (70 μm). The samples were stained with anti-PercpCy5.5-CD3, anti-FTTC-CD4, anti-APC-CD8, anti-PE-CD69 antibodies (100 ×) and then analyzed by FCM. The data were performed with Flow Jo software.

### Statistics analysis

The results were expressed as the mean ± s.d., and data analyses were performed using the Prism software (Version 8.0, GraphPad Software). Statistical analysis was performed via one-way ANOVA. The $p$ value was denoted by * for $p < 0.05$, ** for $p < 0.01$, *** for $p < 0.001$ and **** for $p < 0.0001$.

### Reporting summary

Further information on research design is available in the Nature Portfolio Reporting Summary linked to this article.

## Data availability

The data that support the findings of this study are available within the article and its Supplementary Information files. Data generated in this study are provided in the Source Data file. Source data are provided with this paper.

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

## Acknowledgements

This work was supported by the National Natural Science Foundation of China (82172094, 22205151, and 32201148), Funds of Sichuan Province for Distinguished Young Scholar (2021JDJQ0037), and the China Postdoc Innovation Talent Support program (BX20190223).

## Author contributions

N.W., C.L., and C.Y.G. designed the experiments. N.W. and C.L. contributed to the execution of experiments and analysis of the data. Y.J.L. and D.X.H. performed the in vitro transfection and characterization of the nanoparticles. X.Y.Wu, X.R.K., and X.Y.Wang. established the tumor model and performed the in vivo anti-tumor effect. C.Y.G. and Q.J.W. supervised the project and provided research guidance.

## Competing interests

The authors declare no competing interests.
