## [Peer Review File · Nature Communications]

A cooperative Nano-CRISPR scaffold potentiates immunotherapy via activation of tumour-intrinsic pyroptosisREVIEWER COMMENTS

Reviewer #1 (Remarks to the Author):

In this paper, Ning Wang and colleagues constructed a cooperative NanoCRISPR scaffold (Nano-CD) for intracellular pyroptosis-based immunotherapy. The Nano-CD is fabricated by coating a versatile copolymer on amino acid modified cationic core for CRISPR/dCas9 and cisplatin co-delivery. By fusing the nuclease-inactivated Cas9 with transcription activators, CRISPR/dCas9 can use the own genome of tumor cells to produce therapeutic proteins with bioactivity, providing a precise and self-supply modality to yield GSDME during pyroptosis process. Cisplatin and CRISPR/dCas9 plasmid can be release from Nano-CD in the acidic condition in tumour. High level of GSDME protein expression and strong tumor pyroptosis was detected by treating tumour cells with Nano-CD, which further reversing the immunosuppressive TME and boosting the antitumor immune cascade as positive feedback. When combined with immune checkpoint blockade therapy, Nano-CD was able to inhibit recurrence and metastasis of malignant melanoma which exhibited strong systemic antitumor immune responses and durable immune memory effect. This is a novel design, the paper is well written and organized. An impressive amount of data was collected and the in vivo experiments showed great tumour inhibition. This paper can be accepted once they addressed some minor concerns below:

1. Cisplatin is reported to induce DNA cross-linking to induce cancer cell death, did the authors found any effect of plasmid DNA damage when formulating the Nano-CD? This should be evaluated.
2. The authors should clarify why they choose plasmid DNA but not mRNA to encode the CRISPR/dCas9. Usually the immunogenicity of mRNA is much lower than plasmid DNA and can achieve higher protein expression in vivo.
3. How the Nano-CD is taken up by the cells is unknown. Intracellular distribution and endosomal escape of the cargo material using confocal should be investigated. For example, Gong N, Zhang Y, Teng X, et al. Proton-driven transformable nanovaccine for cancer immunotherapy. *Nature nanotechnology*, 2020, 15(12): 1053-1064.
4. The biodistribution data in Figure 3e-f is very impressive. Usually nanoparticles mainly locate in liver after they are i.v. injected, how these nanoparticles can achieve such tumor cell-specific delivery is unknown.
5. The stability of the Nano-CD in both PBS and serum should be evaluated.
6. Flow gate strategies for Figure 4d-e, Figure5e-I, and Figure6 e-g should be provided in SI.
7. Color scale in figure 3 f is missing.
8. How statistical analysis in The figures 1-6 is unknown, this information should be indicated in the figure legends.

Reviewer #2 (Remarks to the Author):

A Cooperative NanoCRISPR Scaffold Potentiated Immunotherapy via Self-supply of Pyroptotic Protein

Recommendation: publish after major revision.

Comments:

The clinical application of bioactive protein has been substantially hampered by extremely high requirements along with poor bioavailability and complicated preparation. The authors report a cooperative NanoCRISPR scaffold (Nano-CD) for enhanced immunotherapy via bioactive protein self-supply. The experiments were performed carefully, various experimental techniques have been used.

However, the authors should address these issues before considering publishing in *Nature Communications*.

In line 86 and Supplementary Fig. 1, what is the modification efficiency of amino acid grafted from the PEI 1.8k? Does the modification efficiency affect the self-assembly of Nano-CD?

The data shown in Supplementary Fig. 2 could not fully quantitatively prove the modification of TAT and Pt. In addition to, NMR and FT-IR, the author should consider Mass, ICP-MS et al. further, what is the modification efficiency of TAT and Pt? Does the modification efficiency affect the self-assembly of Nano-CD?

In Supplementary Fig. 3. "Cisplatin release behavior under pH 5.0 or pH 7.4 at different time points by HPLC". The author should show the HPLC or LC-MS elution peaks against retention time.

In Supplementary Fig. 4, the statistical difference analysis between experimental and control groups should be added.

In Supplementary Fig. 5, all the CLSM images should be attached at least one scale bar.

The resolution of the CLSM images is not high enough.

In Supplementary Fig. 2. The chemical structure of PAA was shown incorrectly.

The author claim that the Nano-CD will disassemble in pH 5.0 and release the GSDME-cas9 core, however, the pKa of acrylic acid is about 5, Why the self-assembly will not occur due to the protonation of polyacrylic acid?

In Fig. 2, the resolution of the images is not high enough to prove the occurring of pyroptosis.

Moreover, the number of cells is not enough to prove the universality of the phenomenon.

Reviewer #3 (Remarks to the Author):

In this manuscript, Gong et al developed a cooperative NanoCRISPR scaffold for intracellular pyroptosis-based immunotherapy via self-supply of therapeutic protein GSDME, wherein the CRISPR/dCas9 acts directly on tumor cells and thus minimizes immune-related adverse effects. In fact, pH-responsive nanocarriers constructed by the self-assembly of copolymers have been widely adopted in the past decade. Furthermore, the role of GSDME in inhibiting tumor growth has also been reported in previous works (Nature Reviews Immunology, 2020, 20, 274; Biomaterials, 2020, 254, 120142). Overall, this is well-organized work. However, the manuscript focused on the development of co-delivery nanocarriers and their potential applications in immune therapeutics, which was interesting but lack of novelty. It might be suitable to be published in a more specialized journal focusing on biomedical applications.

1. Initial leakage of CRISPR/dCas9 from the polymer micelles at different solution pH (4.0/5.0/6.0/7.0/8.0) should be provided.

2. The size of micelles ranges from 37nm to 342 nm. The fabrication process of micelles should be optimized, which has an important impact on protein loading efficiency

3. Some related references about protein release may be helpful to improve the introduction section of this manuscript.

Reviewer #1 (Remarks to the Author):

In this paper, Ning Wang and colleagues constructed a cooperative NanoCRISPR scaffold (Nano-CD) for intracellular pyroptosis-based immunotherapy. The Nano-CD is fabricated by coating a versatile copolymer on amino acid modified cationic core for CRISPR/dCas9 and cisplatin co-delivery. By fusing the nuclease-inactivated Cas9 with transcription activators, CRISPR/dCas9 can use the own genome of tumor cells to produce therapeutic proteins with bioactivity, providing a precise and self-supply modality to yield GSDME during pyroptosis process. Cisplatin and CRISPR/dCas9 plasmid can be release from Nano-CD in the acidic condition in tumour. high level of GSDME protein expression and strong tumor pyroptosis was detected by treating tumour cells with Nano-CD, which further reversing the immunosuppressive TME and boosting the antitumor immune cascade as positive feedback. When combined with immune checkpoint blockade therapy, Nano-CD was able to inhibit recurrence and metastasis of malignant melanoma which exhibited strong systemic antitumor immune responses and durable immune memory effect. This is a novel design, the paper is well written and organized. An impressive amount of data was collected and the in vivo experiments showed great tumour inhibition. This paper can be accepted once they addressed some minor concerns below:

Our response: We appreciate the positive comments of the reviewer.

1. Cisplatin is reported to induce DNA cross-linking to induce cancer cell death, did the authors found any effect of plasmid DNA damage when formulating the Nano-CD? This should be evaluated.

Our response: Thank you very much for your valuable comments. The anti-tumor

mechanism of cisplatin is to cause DNA cross-linking, which impairs its replication and thus causes cell death. Therefore, that plasmid is not damaged by cisplatin is crucial for Nano-CD to initiate tumor pyroptosis. We evaluated the effect of cisplatin on plasmid from the following three aspects.

(1) The structure of Nano-CD. The CRISPR/dCas9 plasmid and PEI_{PT} form the core by electrostatic adsorption, while cisplatin is modified on the surface of the polyacrylic acid corona. According to the design, the plasmid and cisplatin are spatially in the inner and outer layers of Nano-CD, respectively. Therefore, we avoided the impact of cisplatin on plasmid function by spatial structure design.

(2) The amplification of plasmids. When Nano-CD is depolymerized intracellularly, cisplatin and plasmid are released into the cell to initiate pyroptosis. In order to evaluate whether cisplatin will cause plasmid damage at this stage, we evaluated the amplification of CRISPR/dCas9 plasmid within Nano-CD. If cisplatin caused DNA cross-linking, this would inhibit plasmid amplification. First, we incubated Nano-CD in an acidic condition (pH 5.0) which was simulated the lysosome environment for 48 hours. After treatment, the plasmid within Nano-CD was purified and transformed into *Escherichia coli*. The results showed that the transformed *E. coli* was able to form clones in ampicillin-resistant agar plates, indicating that the CRISPR/dCas9 plasmid with ampicillin resistance was successfully expressed in *E. coli*. Moreover, the number of clones formed by *E. coli* in the Nano-CD group was not statistically different from the number of clones formed in the pure plasmid group (**Fig. R1**). This result indicated that plasmid replication was not impaired obviously by cisplatin in Nano-CD system.

(3) The function of protein self-supply. The realization of protein self-supply is the manifestation that cisplatin does not damage the function of plasmid. Therefore, we

transfected B16F10 cells using Nano-CD and nanoparticles without modified cisplatin, respectively. As shown in the **Fig. R2**, the activation of GSDME was not significantly different in the two groups. From the above results, we did not observe an obvious damage effect of cisplatin on the function of CRISPR/dCas9 plasmid.

Fig. R1. Colony formation experiment of Escherichia coli.

Fig. R2. (g, h) CC3, GSDME, GSDME-N, β -actin expression with different treatments by Western Blots assay. This figure was included as **Fig. 1g** and **h** in the revised manuscript.

2. The authors should clarify why they choose plasmid DNA but not mRNA to encode the CRISPR/dCas9. Usually the immunogenicity of mRNA is much lower than plasmid DNA and can achieve higher protein expression in vivo.

Our response: We sincerely appreciate the insightful comments of the reviewer. Exactly, mRNA does not need to enter the nucleus to be transcribed. In addition, recent studies have also found that the addition of modified nucleotides can reduce the immunogenicity of mRNA and improve the translation efficiency of proteins (Hou X et al., Nature Reviews Materials 2021; 6(12): 1078-1094). Benefit from the advantages, there is no doubt that mRNA has the priority for clinical application.

In this study, we chose plasmid instead of mRNA, mainly considering the efficiency of protein self-supply. The CRISPR/dCas9 system consists of dCas9 and sgRNA. The two components can be constructed in a single plasmid system, while when using mRNA, CRISPR/dCas9 will be split into two separate RNAs with a large difference in length (dCas9 mRNA: ~5000 nt; sgRNA: ~300 nt). In our pre-experiment, we found that Nano-CD loaded with CRISPR/dCas9 plasmid had a higher efficiency of GSDME protein self-supply compared to loading two independent RNAs. Based on the results, we speculate that there are two important issues at this stage using mRNA: (1) Determination of the ratio of dCas9 mRNA and sgRNA for the optimal protein self-provisioning efficiency. (2) How to regulate the efficiency of Nano-CD to load different lengths of mRNA simultaneously. In addition, DNA is more stable than mRNA, which may make the protein self-supply of CRISPR/dCas9 more durable. Therefore, we chose CRISPR/dCas9 plasmid for this study.

Considering the potential of mRNA in clinical applications, we are also trying to overcome the above-mentioned challenges of mRNA application in protein self-supplying system, and we hope to obtain positive results in the future.

3. How the Nano-CD is taken up by the cells is unknown. Intracellular distribution and endosomal escape of the cargo material using confocal should be investigated. For example, Gong N, Zhang Y, Teng X, et al. Proton-driven transformable nanovaccine for cancer immunotherapy. *Nature nanotechnology*, 2020, 15(12): 1053-1064.

Our response: Thank you very much for the valuable comments. According to the reference provided by the reviewer, the cell uptake and intracellular delivery behavior of Nano-CD was studied (**Fig. R3a and b**). The cell uptake efficiency of Nano-CD increased ~1-fold than that of PEI 25K. Meanwhile, the intracellular behaviors of Nano-CD were explored with confocal laser scanning microscope (CLSM). The CLSM images indicated that the YOYO-1 labelled Nano-CD (green) began to accumulate to the cell membrane at 1 h. After incubated for 2 h, the green signal trafficked to the lysosome and enriched in lysosome at 4 h. Subsequently, the green signal almost completely overlapped with blue (nucleus) at 8 h, which implied that Nano-CD escaped from lysosome and successfully released the plasmids owing to the layer degradation and “proton sponge” effect of polyethyleneimine (**Fig. R3c**). The excellent property of Nano-CD scaffold would consequently contribute to the realization of self-supply of protein in tumor cells. Meanwhile, the images with high resolution have been added to the revised manuscript (**Supplementary Fig. 9**).

Fig. R3. Cell uptake analysis. **a, b**, Representative cell uptake of different agents by FCM. **c**, Intracellular tracking of Nano-CD in B16F10 cells. This figure was included as **Supplementary Fig. 9** in the revised Supplementary Information.

4. The biodistribution data in Figure 3e-f is very impressive. Usually nanoparticles mainly locate in liver after they are i.v. injected, how these nanoparticles can achieve

such tumor cell-specific delivery is unknown.

Our response: We appreciate the insightful comments of the reviewer. Generally, nanoparticles tend to locate in liver after intravenously injection. The specific accumulation of Nano-CD in tumors is mainly due to the following reasons:

(1) The enhanced permeability and retention (EPR) effect of tumors enhanced the tumor-specific accumulation of Nano-CD with diameter of about 160 nm.

(2) The majority of plasma proteins have an isoelectric point at pH 4 to 6, which results in proteins in plasma that are mostly negatively charged. Similarly, Nano-CD is negatively charged, a feature that reduces the binding of nanoparticles to plasma proteins and avoids the capture of Nano-CD by the reticuloendothelial (RES) system.

(3) Previous studies have shown that particles with a size of 100 to 200 nm have less accumulation in the liver. Nano-CD has a particle size of about 160 nm, which may also account for the less distribution in the liver (Lin D et al., *Biochim Biophys Acta* 1992; 1104(1):95-101; Moreira JN et al., *Biochim Biophys Acta* 2001; 1515(2):167-176; Torchilin VP et al., *Proc Natl Acad Sci USA* 2003; 100(10):6039-6044;).

In summary, there are two reasons why Nano-CD exhibits specific accumulation in tumor. EPR effect enhanced accumulate of Nano-CD in tumors, while the particle size and electrical properties of Nano-CD reduced its distribution in the liver.

5. The stability of the Nano-CD in both PBS and serum should be evaluated.

Our response: Thank you very much for your valuable comments. Your suggestion was very helpful for improving our study. According to your comment, we evaluated the stability of Nano-CD in a variety of mediums and added the data to the supporting information. As shown in **Fig. R4**, the particle size of Nano-CD was changed from 157 nm to 160 nm in PBS form 24h to 48 h, and the zeta was changed from -5.3 mV to -4.8

mV in PBS from 24h to 48 h. In addition, in medium containing 5% serum, the size of Nano-CD was changed from 168 nm to 179 nm from 24h to 48 h, and the zeta was changed from -4.2 mV to -3.9 mV from 24h to 48 h. Furthermore, we also performed an agarose gel retardation assay to determine DNA condensation capacities of Nano-CD at 24 h or 48 h in PBS and medium with 5% serum. The results showed that Nano-CD could retard CRISPR/dCas9 plasmid mobility effectively in all the conditions.

Fig. R4. The stability of Nano-CD in PBS or 5% serum for 24h and 48 h.

a,b, The particle size and zeta potential of Nano-CD in PBS or 5% serum for 24 h and 48 h. **c**, Agarose gel electrophoresis of Nano-CD in PBS or 5% serum for 24 h and 48 h. This figure was included as **Supplementary Fig. 6** in the revised Supplementary Information.

6. Flow gate strategies for Figure 4d-e, Figure 5e-I, and Figure 6 e-g should be provided in SI.

Our response: Thank you very much for your valuable comments. The flow gate strategies for the analysis of immune microenvironment were provided in supplementary **Fig. R5-13**. Meanwhile, the strategies have been added to the revised Supplementary Information.

Fig. R5. Representative scatter plots and gating information derived from analysis of CD4⁺ T cells in tumor tissue. This figure was included as **Supplementary Fig. 19** in the revised Supplementary Information.

Fig. R6. Representative scatter plots and gating information derived from analysis of CD8⁺ T cells in tumor tissue. This figure was included as **Supplementary Fig. 20** in the revised Supplementary Information.

Fig. R7. Representative scatter plots and gating information derived from analysis of CD206⁺ macrophages in tumor tissue. This figure was included as **Supplementary Fig. 21** in the revised Supplementary Information.

Fig. R8. Representative scatter plots and gating information derived from analysis of DCs in tumor tissue. This figure was included as **Supplementary Fig. 22** in the revised Supplementary Information.

Fig. R9. Representative scatter plots and gating information derived from analysis of T_{EM} cells in tumor tissue. This figure was included as **Supplementary Fig. 23** in the revised Supplementary Information.

Fig. R10. Representative scatter plots and gating information derived from analysis of CD69⁺ T cells in tumor tissue. This figure was included as **Supplementary Fig. 26** in the revised Supplementary Information.

Fig. R11. Representative scatter plots and gating information derived from analysis of Treg in tumor tissue. This figure was included as **Supplementary Fig. 27** in the revised Supplementary Information.

Fig. R12. Representative scatter plots and gating information derived from analysis of CD4⁺ and CD8⁺ T cells in spleens. This figure was included as **Supplementary Fig. 28** in the revised Supplementary Information.

Fig. R13. Representative scatter plots and gating information derived from analysis of CD69⁺ T cells in spleens. This figure was included as **Supplementary Fig. 30** in the revised Supplementary Information.

7. Color scale in figure 3 f is missing.

Our response: Thank you very much for the valuable comments. The scale bar has been added to **Fig. 3f** in the revised manuscript.

Fig. R14. Ex vivo IVIS imaging post i.v. injection of Nano-CD at 8-72 h (n=5).

This figure was included as **Fig. 3f** in the revised manuscript.

8. How statistical analysis in The figures 1-6 is unknown, this information should be indicated in the figure legends.

Our response: We sincerely appreciate the insightful comments of the reviewer. According to the suggestion, the statistical analysis in **Fig. 1-6** were indicated in the figure legends. Meanwhile, the corresponding descriptions have been added to the revised manuscript.

Fig. 1. Characterization of Nano-CD nanoparticles. **a.** Schematic illustration of the preparation of CD particles. **b,** TEM images of Nano-CD in pH 7.4 or pH 5.0. **c,** Particle size and zeta potential (**d**) of Nano-CD in pH 7.4 or pH 5.0. **e,** Agarose gel electrophoresis of pDNA in PEI_{PT} and Nano-CD (Lane 1, DNA ladder; lane 2, naked plasmid; lane 3–8, PEI_{PT}@pDNA at mass ratios of 1 : 1, 2 : 1, 5 : 1, 10 : 1, 15 : 1 and 20 : 1, respectively; lane 9, Nano-dCas9 (20 : 5 : 1); lane 10, Nano-CD (20 : 5 : 1) and lane 11, Nano-CD (20 : 5 : 1) in pH 5.0). **f,** qPCR analysis for sgRNA screening and GSDME expression. (**g, h**) CC3, GSDME, GSDME-N, β -actin expression with different treatments by Western Blots assay. (**i**) FCM assay of apoptosis by Annexin V/PI staining (n = 4). Data are from one representative of three independent experiments. Data are presented as the mean \pm s.d. (n = 3 biological replicates per group) and statistically analyzed using one-way ANOVA and Tukey's tests. (* $p < 0.05$, ** $p < 0.01$, *** $p < 0.001$).

Fig. 2. Representative characterization of pyroptosis. **a,** The morphological images of pyroptosis by CLSM (scale bar, 10 μ m). **b,** Observation of ruptured cell membrane by CLSM (cell membrane: red; Nucleus: blue, scale bar, 20 μ m). **c,** The extracellular secretion of ATP, LDH and IL-1 β . **d,** CRT expression analysis by CLSM (n=4). **e,** CLSM observed the elimination of HMGB1 in B16F10 cells (scale bar, 20

μm). **f**, Quantitative examination of uric acid ($n = 4$). **g**, HSP-70 and β -actin expression with different treatments by Western Blots assay. **h**, The secretion of HMGB1 in cell supernatant by ELISA ($n = 4$). Data are presented as the mean \pm s.d. ($n = 4$ biological replicates per group) and statistically analyzed using one-way ANOVA and Tukey's tests. ($*p < 0.05$, $**p < 0.01$, $***p < 0.001$).

Fig. 3. Immune stimulation of DCs by Nano-CD *in vitro*. **a**, Illustration of the experimental procedure *in vitro*. **b**, The mechanism illustration of DCs maturation. **c**, ELISA assay for TNF- α , IL-12p40 and IFN- γ proinflammatory cytokines. Data are presented as the mean \pm s.d. ($n = 4$ biological replicates per group) and statistically analyzed using one-way ANOVA and Tukey's tests. **d**, Profile of DCs maturation by FCM. Data are presented as the mean \pm s.d. ($n = 3$ biological replicates per group) and statistically analyzed using one-way ANOVA and Tukey's tests. **e-g**, Real-time *in vivo* and *ex vivo* IVIS imaging of Nano-CD after intravenously post *i.v.* injection of Nano-CD at 8 h, 12 h, 24 h, 48 h and 72 h ($n = 5$). The fluorescence intensity of tumor tissues was also quantified. Data are presented as the mean \pm s.d. ($n = 6$ biologically independent animals). ($*p < 0.05$, $**p < 0.01$, $***p < 0.001$).

Fig. 4. Anti-tumor effect of Nano-CD against primary melanoma model. **a**, Time schedule of the treatment. **b**, Individual tumor growth curve and average tumor curve during treatment ($n = 5$). **c**, Survival rate per group. Statistical significance was calculated using a two-way ANOVA test for tumor growth data and log-rank test for survival curves. **d** and **e**, The percentage of CD8 $^+$ T cells, CD4 $^+$ T cells, CD206 $^+$ macrophages, DCs and Foxp3 $^+$ Treg cells in tumors ($n = 4$). Data are presented as the mean \pm s.d. ($n = 4$ biological replicates per group) and statistically analyzed using one-way ANOVA and Tukey's tests. ($*p < 0.05$, $**p < 0.01$, $***p < 0.001$).

Fig. 5. Anti-tumor effect of Nano-CD against recurrence melanoma model. **a**, Scheme of mice treated with different formulations. **b**, Images of tumors from different group after treatment (n = 5), tumor volume, tumor weight (**c**) and survival curve (**d**) of tumor-bearing mice with different treatments. Statistical significance was calculated using a two-way ANOVA test for tumor growth data and log-rank test for survival curves. **e-i**, Profiles and percentages of CD4⁺ T cells, CD8⁺ T cells, CD69⁺ activated T cells, CD44⁺CD62L⁻ T_{EM} cells in tumors and CD4⁺ T cells/CD8⁺ T cells in spleens by FCM analysis. **j**, Concentration of IL-12, IL-10 and TGF- β cytokines in serum after different treatments. Data are presented as the mean \pm s.d. (n = 4 biological replicates per group) and statistically analyzed using one-way ANOVA and Tukey's tests. (* p < 0.05, ** p < 0.01, *** p < 0.001).

Fig. 6. Assessment of therapeutic efficacy of Nano-CD in melanoma pulmonary metastasis model. **a**, Schematic illustration of administration schedule *in vivo*. **b**, The number of nodules on the lungs was counted and the lungs from each group was pictured at day 20 (n = 6). **c**, Survival curve of mice with different treatments. Statistical significance was calculated using a two-way ANOVA test for tumor growth data and log-rank test for survival curves. **d**, Representative images of lungs at the end of the treatment by H&E assay. **e** and **f**, CD4⁺ T cells/CD8⁺ T cells and CD69⁺ activated T cells in spleens of each group. **g**, Level of TGF- β , IL-10, IL-12 and IFN- γ cytokines in serum per group. Data are presented as the mean \pm s.d. (n = 4 biological replicates per group) and statistically analyzed using one-way ANOVA and Tukey's tests. (* p < 0.05, ** p < 0.01, *** p < 0.001).

Reviewer #2 (Remarks to the Author):

The clinical application of bioactive protein has been substantially hampered by extremely high requirements along with poor bioavailability and complicated preparation. The authors report a cooperative NanoCRISPR scaffold (Nano-CD) for enhanced immunotherapy via bioactive protein self-supply. The experiments were performed carefully, various experimental techniques have been used. However, the authors should address these issues before considering publishing in Nature Communications.

Our response: We appreciate the positive comments of the reviewer.

1. In line 86 and Supplementary Fig. 1, what is the modification efficiency of amino acid grafted from the PEI 1.8k? Does the modification efficiency affect the self-assembly of Nano-CD?

Our response: Thanks to the reviewer for the professional comment. In this study, we modified PEI 1.8K with amino acid ranging from 1.25% to 35.63% to prepare Nano-CD system, which was shown in **Table. R1**. We found that different modification efficiencies of PEI 1.8K did not affect the self-assembly of Nano-CD, but dramatically affected their transfection efficiency. The transfection efficiency of Nano-CD was the highest when the modification efficiency was 34.38%. When the modification efficiency was 35.63%, the transfection efficiency of PEI_{PT} was not improved but the yield was reduced by 80%. Since the transfection efficiency determines the protein self-supply function of Nano-CD, we choose PEI 1.8K with an amino acid modification efficiency of 34.38% for the preparation of Nano-CD in our study.

Table. R1. Modification efficiency of amino acid on PEI_{PTn}

Polymer	Number of Phe + Tyr	X	Y
PEI _{PT2}	1 + 1	0.2	1.25%
PEI _{PT4}	2 + 2	0.9	5.62%
PEI _{PT8}	4 + 4	2.3	14.34%
PEI _{PT16}	8 + 8	3.6	22.50%
PEI _{PT20}	10 + 10	5.5	34.38%
PEI _{PT24}	12 + 12	5.7	35.63%

X is the average number of Phe and Tyr on each PEI 1.8K molecular.

Y is the percentage of amino groups on prepared PEI_{PTn}.

This **Table. R1** was included as **Supplementary Table. 1** in the revised Supplementary Information.

2. The data shown in Supplementary Fig. 2 could not fully quantitatively prove the modification of TAT and Pt. In addition to, NMR and FT-IR, the author should consider Mass, ICP-MS et al. further, what is the modification efficiency of TAT and Pt? Does the modification efficiency affect the self-assembly of Nano-CD?

Our response: Thanks for your valuable comments. Your insightful suggestion was very helpful for improving our study. We performed the analysis of TAT peptide within Nano-CD through Mass (**Fig. R15**), while the analysis of cisplatin was achieved by HPLC (**Fig. R16**). The results of Mass showed that the TAT peptide was successfully modified on the PAA surface with the grafting ration of 5%. The HPLC assay showed that the grafting rate of cisplatin was about 15% (The standard curve: $y=7.527x$, $R^2=0.9995$). Based on this data, the total molecular weight of PCT is calculated to be 6000-7000, which is consistent with the mass spectrum results.

In this study, polyacrylic acid (PAA) has a molecular weight of 2000-3000 which contains about 30-40 carboxyl groups. Considering that the polymer shell is negatively charged in order to self-assemble with the positively charged PEI_{PT}@pDNA by electrostatic adsorption. Therefore, the PAA backbone needs to retain a certain number of carboxyl groups. Meanwhile, to ensure the dose of cisplatin and the modification ratio of TAT in Nano-CD, we screened different grafting ratios of the two reagents and finally reacted half of the carboxyl groups on the surface of PAA. Consequently, Nano-CD has the desired function while the nanoparticle still has a stable morphology.

Fig. R15. Mass spectrum of PCT. This figure was included as **Supplementary Fig. 3** in the revised Supplementary Information.

Fig. R16. HPLC spectrum of PCT. a, The standard curve of cisplatin. **b,** The HPLC spectrum of PCT.

3. In Supplementary Fig. 3. “Cisplatin release behavior under pH 5.0 or pH 7.4 at different time points by HPLC”. The author should show the HPLC or LC-MS elution peaks against retention time.

Our response: We appreciate the insightful comments. According to the suggestion of the reviewer, we showed the HPLC elution peaks against retention time under pH 5.0 and pH 7.4. As shown in **Fig. R17**, PCT showed significant release of cisplatin under pH 5.0 condition, suggesting that PCT is capable of releasing cisplatin in an acidic environment that mimics lysosomes. Meanwhile, the data have been added to **Supplementary Fig. 5**.

Fig. R17. HPLC spectrum of Cisplatin or PCT in pH 7.4 for 24 h or PCT in pH 5.0 for 24 h. This figure was included as Supplementary Fig. 5 in the revised Supplementary Information.

4. In Supplementary Fig. 4, the statistical difference analysis between experimental and control groups should be added.

Our response: Thanks for your valuable comments. The statistical difference analysis was shown in **Fig. R18**. As shown in **Fig. R18a**, the statistical difference between PEI_{PT} and PEI 25K group was $***p < 0.0001$, when the drug concentration was from 18.75 to 300 $\mu\text{g/mL}$. As shown in **Fig. R18b**, there was no statistical difference among the

PAA, PT and PCT group. Meanwhile, the data have been added to **Supplementary Fig.**

4.

Fig. R18. MTT assay of different materials. This figure was included as **Supplementary Fig. 8** in the revised Supplementary Information.

5. In Supplementary Fig. 5, all the CLSM images should be attached at least one scale bar. The resolution of the CLSM images is not high enough.

Our response: Thanks for your valuable comments. Your suggestion is very helpful for improving our study. We have added scale bar (20 μm) in the CLSM image in **Fig. R19 (Supplementary Fig. 9)**. According to the suggestion of the reviewer, we replaced **Supplementary Fig. 9** with high-quality CLSM images.

Fig. R19. Cell uptake analysis. **a, b**, Representative cell uptake of different agents by FCM. **c**, Intracellular tracking of Nano-CD in B16F10 cells. This figure was included as **Supplementary Fig. 9** in the revised Supplementary Information.

6. In Supplementary Fig. 2. The chemical structure of PAA was shown incorrectly.

Our response: We apologize for the mistake, and it has been corrected in the revised manuscript. The correct structure of PAA has been shown in **Fig. R20** and also added to the revised manuscript (**Supplementary Fig. 2**).

Fig. R20. a, Synthesis routines of Shell. **b,** Spectrum of Shell in D₂O. **c,** FTIR of PCT. This figure was included as **Supplementary Fig. 2** in the revised Supplementary Information.

7. The author claim that the Nano-CD will disassemble in pH 5.0 and release the GSDME-cas9 core, however, the pKa of acrylic acid is about 5, Why the self-assembly will not occur due to the protonation of polyacrylic acid?

Our response: Thanks for your valuable comments. The pKa of acrylic acid is about 5.0, so PAA is protonated in pH 5.0, which causes its charge changes from negative to positive. As a consequent, the positively charged PEI_{PT}@pDNA core and the protonated PAA shell (positively charged) repel each other, leading to the disassembly of Nano-CD.

8. In Fig. 2, the resolution of the images is not high enough to prove the occurring of pyroptosis. Moreover, the number of cells is not enough to prove the universality of the phenomenon.

Our response: Thanks for your valuable comments. Your insightful suggestion was very helpful for improving our study. The images we have provided may cause the reader to be misled about the universality of pyroptosis. Therefore, we repeated the experiment and recorded the cell morphology after Nano-CD treatment. The majority of Nano-CD treated B16F10 cells showed evident swelling with large bubbles (Red arrow) from the plasma membrane which is the typical morphology of pyroptosis (**Fig. R21**). This result proves the universality of pyroptosis.

Based on the pore forming mechanism of pyroptosis, the integrity of the cell membrane would be disrupted, followed by the release of cell contents including pro-inflammatory molecules and antigens into the tumor microenvironment. The T11 dye (Red), which could not penetrate the normal cell membrane, successfully stained the proteins in the cytoplasm of the Nano-CD treated cells. The increased red signal implied Nano-CD greatly destroys the integrity of cell membrane by the cooperation of cisplatin

and CRISPR/dCas9 (Fig. 2b). Furthermore, the release of cell contents, including adenosine triphosphate (ATP) and lactate dehydrogenase (LDH) in Nano-CD group were strongly higher than that of Nano-C or Nano-dCas9 treated only (Fig. 2c). Meanwhile, the level of IL-1 β cytokines, a signal molecule in the classical pyrolytic signaling pathway, also increased significantly in Nano-CD group. According to above results, we confirmed that Nano-CD initiated robust pyroptosis via the collaboration of self-supply of GSDME protein and cisplatin. The corresponding descriptions have been added to the revised manuscript.

Fig. R21. The morphological images of pyroptosis by CLSM. This figure was included as **Supplementary Fig. 11** in the revised Supplementary Information.

Reviewer #3 (Remarks to the Author):

In this manuscript, Gong et al developed a cooperative NanoCRISPR scaffold for intracellular pyroptosis-based immunotherapy via self-supply of therapeutic protein GSDME, wherein the CRISPR/dCas9 acts directly on tumor cells and thus minimizes immune-related adverse effects. In fact, pH-responsive nanocarriers constructed by the self-assembly of copolymers have been widely adopted in the past decade. Furthermore, the role of GSDME in inhibiting tumor growth has also been reported in previous works (Nature Reviews Immunology, 2020, 20, 274; Biomaterials, 2020, 254, 120142). Overall, this is well-organized work. However, the manuscript focused on the development of co-delivery nanocarriers and their potential applications in immune therapeutics, which was interesting but lack of novelty. It might be suitable to be published in a more specialized journal focusing on biomedical applications.

Our response: We appreciate the critical comments of the reviewer. We must apologize for causing the misunderstanding of protein delivery. Distinct from the protein delivery, we constructed a nanoplatfrom (Nano-CD) with core-shell structure for self-supply of therapeutic protein without exogenous delivery. In this work, we reported a cooperative NanoCRISPR scaffold (Nano-CD) for enhanced immunotherapy via bioactive protein self-supply. After Nano-CD treatment, unlocking expression of GSDME protein (originally low expressed in tumor cells), cisplatin, and intracellular acidic condition work in a synergetic modality to specifically initiate robust tumor pyroptosis. Subsequently, pyroptosis released tumor associated antigens, and the lytic intracellular contents conferred adjuvant properties, cascade amplifying the antitumor immune responses. Nano-CD treatment efficiently inhibited tumor growth in primary and recurrence melanoma models, and a durable immune memory effect and strong

systemic anti-tumor immune response were provoked when combine with checkpoint blockade.

The wide application of copolymer shows its potential in many fields, especially in anti-tumor therapy. For different purposes, copolymers can achieve versatile functions through novel design by virtue of continuous innovation of structure and materials. Nano-CD is a noncomplex with distinctive properties, which is formed by the multifunctional shell of PAA backbone and amino acid modified cationic core through two-step self-assembly. By modifying cisplatin on the surface of PAA while compressing the plasmid in the core, a cooperative NanoCRISPR scaffold with spatial blocking effect is formed. Rather than delivering two drugs with similar features, the delivery of two agents with different physical and chemical properties, chemical drugs and CRISPR/dCas9 plasmid, poses a huge challenge. The issues include controlling the uniform size of nanoparticles, appropriate drug loading ratio, the way of cooperation between the two drugs, efficient cell internalization, etc. The above mentioned issues were solved in our study.

As demonstrated by the reviewer, the anti-tumor function of GSDME has been reported. In our work, GSDME was selected as the model target of protein self-supply because of its unique biological characteristics and the bottleneck of anti-tumor applications. Firstly, GSDME proteins are not expressed or expressed extremely low in various tumor types. Previous studies mainly induced pyroptosis through GSDME activation by DNA methyltransferase inhibitor (e.g. decitabine). However, decitabine is a pan-regulation drug, and there may be a risk of activating the oncogene while activating the GSDME expression. Secondly, exogenous delivery of GSDME protein instead of decitabine to regulate pyroptosis inevitably requires protein expression and purification which are costly and laborious. Furthermore, the extremely demanding in

terms of bioactivity of proteins also detracts from successfully moving these strategies from the bench to the clinic. Thus, Nano-CD was conducted to surmount the abovementioned issues via GSDME self-supply.

Therefore, the novelty and superior of our Nano-CD system are mainly focusing on the delicate design of cooperative Nano-CD and bioactive protein self-supply strategy enabled by the cooperative NanoCRISPR scaffold, NOT pH-responsive nanocarriers or anti-tumor effects of GSDME protein. To the best of our knowledge, there is still no report of a cooperative NanoCRISPR scaffold for self-supply of therapeutic protein without exogenous delivery for enhanced immunotherapy thus far. Thus, the novelty of our work is high enough, deserving publication on *Nature Communications*.

1. Initial leakage of CRISPR/dCas9 from the polymer micelles at different solution pH (4.0/5.0/6.0/7.0/8.0) should be provided.

Our response: Thanks for your valuable comments. We investigated the initial leakage of CRISPR/dCas9 from the Nano-CD at different pH (4.0/5.0/6.0/7.0/8.0) conditions through agarose gel electrophoresis. As shown in **Fig. R22**, the plasmid remains inside the Nano-CD and is therefore blocked in the sample pore at pH 6.0-8.0, while the plasmids in Nano-CD were significantly released at both pH 4.0 and 5.0. The results indicate that after intravenously injection, Nano-CD can effectively protect the plasmid from degradation in the blood and prolong the circulation time. After uptaken by tumor cells, Nano-CD is able to depolymerize and release plasmids.

Fig. R22. Agarose gel electrophoresis of Nano-CD in different pH (pH 4.0-8.0).

This figure was included as **Supplementary Fig. 7** in the revised Supplementary Information.

2. The size of micelles ranges from 37 nm to 342 nm. The fabrication process of micelles should be optimized, which has an important impact on protein loading efficiency

Our response: Thank you very much for your comments. The Nano-CD is a core-shell structured nanocomplex NOT micelle, which is formed by two-step self-assembly of multifunctional shells of PAA skeleton and amino acid modified cationic cores. The preparation methods between the core-shell structured nanocomplex and micelles are quite different, so the PDIs and sizes of the two agents are distinct (Cabral H et al., Chem Rev 2018; Tiancong Zhao et al., Nature Reviews Materials 2019; Lu Y et al., Nat Biomed Eng 2018; Yin H et al., Nat Rev Gene 2014; Lanfang Ren et al., Angew Chem Int Ed Engl 2020; Jun Xu et al., Nat Nanotechnol 2020). Exactly, the optimized particle size is critical to the therapeutic performance and clinical translation of Nano-CD. According to the constructive suggestions of the reviewer, we strictly control the

preparation processes including optimizing the reagent purity, plasmid concentration, incubation time and environmental cleanliness in each step of the procedure. Based on the updated procedure, the morphology and PDI of Nano-CD have been improved as shown below (**Fig. R23**).

Fig. R23. Particle size of Nano-CD.

3. Some related references about protein release may be helpful to improve the introduction section of this manuscript.

Our response: Thanks for your valuable comments. The related references about protein release have been added to the revised manuscript (Ref. 17-19 in the revised manuscript).

REVIEWERS' COMMENTS

Reviewer #1 (Remarks to the Author):

The revised manuscript adds substantial new data further confirming key points of the original submission. I appreciate the effort made by the authors to thoroughly respond to the reviewer comments. I find the revised manuscript substantially improved.

Reviewer #2 (Remarks to the Author):

The author has successfully addressed all the concerns. The reviewer recommends accepting this work without further revision.

Reviewer #3 (Remarks to the Author):

I suggest to accept the revised manuscript.

Response to the reviewers' comments:

Reviewer #1 (Remarks to the Author):

The revised manuscript adds substantial new data further confirming key points of the original submission. I appreciate the effort made by the authors to thoroughly respond to the reviewer comments. I find the revised manuscript substantially improved.

Our response: We appreciate the positive comments of the reviewer.

Reviewer #2 (Remarks to the Author):

The author has successfully addressed all the concerns. The reviewer recommends accepting this work without further revision.

Our response: We appreciate the positive comments of the reviewer.

Reviewer #3 (Remarks to the Author):

I suggest to accept the revised manuscript.

Our response: We appreciate the positive comments of the reviewer.